# LabUtopia: High-Fidelity Simulation and Hierarchical Benchmark for Scientific Embodied Agents

**Rui Li**[1,5*]   **Zixuan Hu**[2*]   **Wenxi Qu**[1*]   **Jinouwen Zhang**[1]   **Zhenfei Yin**[1,3]   **Sha Zhang**[1,6]
**Xuantuo Huang**[2]   **Hanqing Wang**[1]   **Tai Wang**[1]   **Jiangmiao Pang**[1]   **Wanli Ouyang**[1,4]
**Lei Bai**[1]   **Wangmeng Zuo**[5]   **Ling-Yu Duan**[2†]   **Dongzhan Zhou**[1†]   **Shixiang Tang**[1,4†]
[1]Shanghai AI Laboratory   [2]Peking University   [3]Oxford [4]The Chinese University of Hong Kong
[5]Harbin Institute of Technology   [6]University of Science and Technology of China

## Abstract

Scientific embodied agents play a crucial role in modern laboratories by automating complex experimental workflows. Compared to typical household environments, laboratory settings impose significantly higher demands on perception of physical-chemical transformations and long-horizon planning, making them an ideal testbed for advancing embodied intelligence. However, its development has been long hampered by the lack of suitable simulator and benchmarks. In this paper, we address this gap by introducing **LabUtopia**, a comprehensive simulation and benchmarking suite designed to facilitate the development of generalizable, reasoning-capable embodied agents in laboratory settings. Specifically, it integrates i) LabSim, a high-fidelity simulator supporting multi-physics and chemically meaningful interactions; ii) LabScene, a scalable procedural generator for diverse scientific scenes; and iii) LabBench, a hierarchical benchmark spanning five levels of complexity from atomic actions to long-horizon mobile manipulation. LabUtopia supports 30 distinct tasks and includes more than 200 scene and instrument assets, enabling large-scale training and principled evaluation in high-complexity environments. We demonstrate that LabUtopia offers a powerful platform for advancing the integration of perception, planning, and control in scientific-purpose agents and provides a rigorous testbed for exploring the practical capabilities and generalization limits of embodied intelligence in future research. Project web page: the github URL.

## 1   Introduction

Scientific breakthroughs play a foundational role in advancing human knowledge [50], driving technological innovation, and improving societal well-being [3]. However, the traditional paradigm of natural science research remains slow and labor-intensive [51], where countless experiments must be performed by skilled researchers to reach meaningful insights [56, 43, 46, 5]. These limitations constrain the overall pace and scalability of scientific discovery. Therefore, automated laboratories [11, 30] have emerged as a promising alternative, aiming at developing intelligent agents that autonomously design and execute complex experiments through adaptive workflows. By reducing human workload, such agents enable scalable, reproducible, and around-the-clock experimentation, significantly increasing research throughput [24]. Nevertheless, building effective scientific agents remains challenging, particularly due to the high cost of data collection and the difficulty of generalizing across diverse hardware platforms [42].

A promising approach to address the above issues is the sim-to-real framework [63], where agents are first trained within realistic simulations before being deployed in real-world laboratory settings. This

---

*Equal contribution.    † Corresponding author.

39th Conference on Neural Information Processing Systems (NeurIPS 2025) Track on Datasets and Benchmarks.

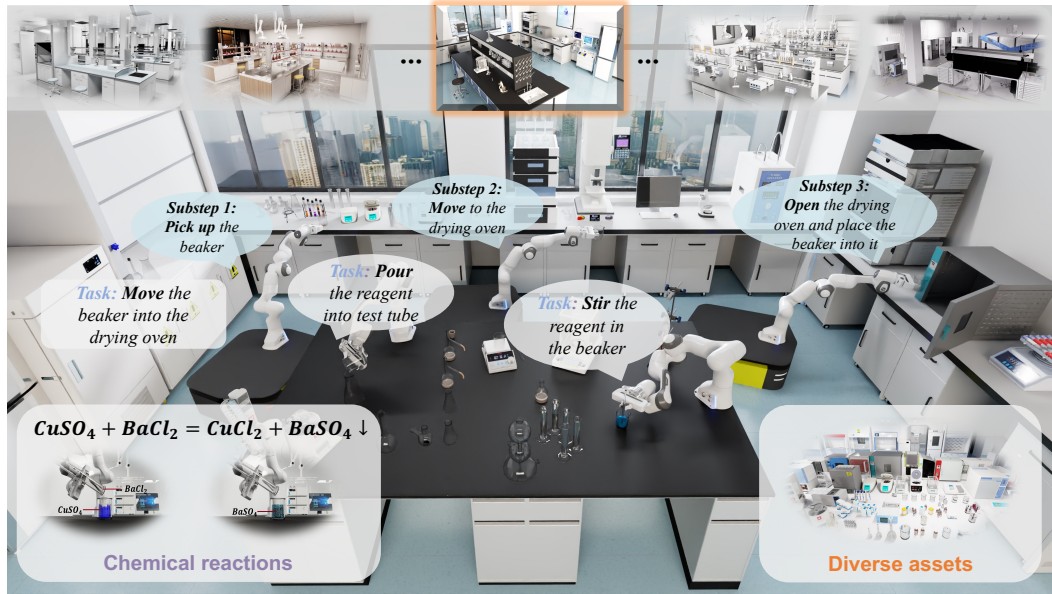

Figure 1: The **LabUtopia** simulation environment and benchmark for developing scientific embodied agents in automated laboratories. LabUtopia supports **chemical reaction** modeling and provides **diverse laboratory assets**, forming a high-fidelity testbed for **tasks of varying difficulty**—from atomic actions to long-horizon action sequences involving both manipulation and navigation.

paradigm enables cost-effective and safe training while maintaining the potential for generalization to physical environments [25, 39, 58]. However, current established simulators predominantly focus on household environments [35, 47, 66, 61, 32, 65] and fail to adequately address the specific challenges of scientific experimentation. As discussed in Table 1, they exhibit three fundamental limitations: (1) the inability to model chemical dynamics, such as product formation or color changes, which are essential for accurate perception and reasoning in lab tasks; (2) limited diversity and semantic richness in their asset libraries, which restricts the faithful representation of heterogeneous laboratory environments in real world; and (3) the lack of comprehensive evaluation protocols, particularly those that span from fine-grained atomic actions to complex, long-horizon experimental procedures.

To address these challenges, we present **LabUtopia**, a comprehensive simulation and benchmarking suite tailored for scientific laboratory contexts. LabUtopia integrates a diverse set of functional assets, a hierarchical task taxonomy, and a high-fidelity simulation engine capable of modeling rigid, deformable, and fluid objects, as well as simulating both physical and chemical processes. Our goal is to provide a scalable, versatile platform for training and evaluating agents' perception, planning, and control skills under high task complexity and diverse environments, advancing the role of embodied intelligence in accelerating scientific discovery. The key designs and innovations include:

(1) *LabSim* is a high-fidelity simulation environment built on Isaac Sim, enhanced with a chemical engine that models reaction-driven transformations (e.g., color change, product generation) by combining a curated substance database with the reasoning model. Extending beyond conventional physical dynamics, LabSim supports a wide range of chemical reactions, enabling precise and visually grounded simulation of laboratory phenomena.

(2) *LabScene* is a procedural generation pipeline that synthesizes diverse, physically plausible 3D laboratory scenes aligned with real-world configurations. Built upon a curated set of expert-verified assets, LabScene employs a hybrid layout strategy that combines grid stochastic sampling with constraint-aware search, enabling scalable environment creation for scientific embodied tasks.

(3) *LabBench* is a hierarchical benchmark, featuring a five-level task structure that spans from atomic object manipulations to long-horizon missions requiring integrated navigation and manipulation. Together, these components establish a rigorous and scalable testbed for developing and evaluating scientific-purpose embodied agents under rich physical constraints and procedural complexity.

To provide an in-depth analysis of embodied agents in scientific laboratory settings, we complement prior research with an evaluation that targets the agent's ability to conduct realistic experimental

procedures that involve accurate material recognition, multi-step task planning, and precise instrument control. Powered by our LabScene generation pipeline, we construct over 100 diverse and physically plausible lab environments, and evaluate agent performance across 30 tasks of varying complexity in the LabBench benchmark. Through extensive experiments, we show that current state-of-the-art manipulation policy models still struggle with the variability of instrument configurations and accumulated errors in long-horizon task execution, highlighting the need for more specialized solutions in research-oriented embodied AI.

We summarize our main contributions as follows:

- We present LabUtopia, a simulation and benchmarking suite tailored to the unique challenges of laboratory settings. LabUtopia supports complex physical interactions and various chemical reactions across 30 distinct tasks, enabling realistic evaluation of embodied agents in high-fidelity scientific scenarios.

- We provide a high-quality asset set comprising over 100 laboratory scenes and 100 scientific instruments, that have been standardized and filtered by domain experts. Building on this, we design an automated scene generation pipeline to produce diverse, scalable lab environments, supporting both real-world alignment and large-scale training and evaluation.

- We introduce a hierarchical benchmark that spans multiple levels of task complexity, from low-level atomic operations to high-level long-horizon reasoning tasks. This structure enables principled assessment of embodied agents' capabilities and reveals performance bottlenecks across varying levels.

## 2 Related Work

**Automated Laboratories.** Current automated laboratories enhance the efficiency and scalability of experiments in chemistry and materials science while reducing costs by integrating machine learning, robotics and modular platforms. Self-driving laboratories (SDLs) [2] automate repetitive tasks, yet often constrain autonomous experimental design. Systems such as Synbot [21], Chemputer [54], MARS-Chem [11], Artificial Chemist [1], Reactivity Explorer [13], and AI-EDISON [30] have demonstrated remarkable performance in organic synthesis, exploratory chemistry, and nanomaterial optimization, enabling standardized and reproducible experiments and accelerating molecular discovery. However, these systems are limited by predefined protocols, hardware dependencies, insufficient task comprehension, and poor real-time adaptability. As a result, their flexibility and intelligence are constrained, making them less suitable for various experimental tasks. Moreover, most of these systems primarily focus on advancing scientific discovery, with limited attention to the long-term development of intelligent embodied systems, thereby overlooking the potential value of embodiment in scientific research. Therefore, we propose a LabUtopia that offers low-cost, high-efficiency workflows and large-scale data acquisition, enhancing flexibility and data-driven capabilities. This environment aims to support the development of embodied intelligence that is adaptable to a wide range of chemical research scenarios.

**Simulators for Embodied AI.** The rapid development of simulators is progressively transitioning from general-purpose functionality to high-fidelity realism. Certain simulators emphasize versatile capabilities, primarily for modeling interactions and dynamic changes in the physical world, facilitating algorithm validation and training. PyBullet [10], Gazebo [34], and RLBench [29] support real-time physical simulations, including rigid body dynamics and collision detection, while offering diverse sensor emulation and deep learning integration. Conversely, other simulators prioritize high-fidelity scene reconstruction to meet the demands of complex real-world task environments. ARNOLD [19], VLMbench [64], Habitat [49], OmniGibson [37, 53], ManiSkill3 [57], and ClevrSkills [22] focus on language-guided task learning in realistic 3D environments, aiming to advance robotic manipulation and human-robot interaction research. These platforms offer vision-language manipulation benchmarks, open-source frameworks, and human-centric evaluations, supporting rich simulations of daily activities, GPU-accelerated parallel robot simulation, and photorealistic rendering, while investigating compositional reasoning and generalization capabilities. However, existing simulation platforms generally lack specialized modeling for laboratory settings and operations. To address this, we propose LabUtopia based on Isaac Sim [44] tailored for chemical laboratories, enabling embodied agents to perform operational learning, path navigation, and task planning in chemical experimental environments. Integrated with visualized simulations of chemical reaction processes, LabUtopia aims to provide critical support for the advancement of embodied intelligence in experimental sciences.

| Simulator/Benchmark | Simulation | | | Lab Environment | | Task Diversity | | | | |
|---|---|---|---|---|---|---|---|---|---|---|
| | Fluid | Physics | Chemistry | Scene | Object | Multi-action | Composed | Generalization | Long-Horizon | Navigation |
| Alfred [52] | ✗ | ✗ | ✗ | ✗ | ✗ | ✔ | ✔ | ✗ | ✗ | ✔ |
| Behavior [37, 53] | ✔ | ✔ | ✗ | ✗ | ✗ | ✔ | ✔ | ✗ | ✗ | ✔ |
| RLBench [29] | ✗ | ✔ | ✗ | ✗ | ✗ | ✔ | ✗ | ✗ | ✗ | ✗ |
| Ravens [62] | ✗ | ✔ | ✗ | ✗ | ✗ | ✗ | ✔ | ✔ | ✔ | ✗ |
| VLMbench [64] | ✗ | ✔ | ✗ | ✗ | ✗ | ✔ | ✔ | ✔ | ✔ | ✗ |
| Arnold [19] | ✔ | ✔ | ✗ | ✗ | ✗ | ✔ | ✗ | ✔ | ✗ | ✗ |
| VIMA-Bench [31] | ✗ | ✔ | ✗ | ✗ | ✗ | ✗ | ✗ | ✔ | ✔ | ✗ |
| Maniskill3 [57] | ✗ | ✔ | ✗ | ✗ | ✗ | ✔ | ✔ | ✔ | ✗ | ✔ |
| Robofactory [48] | ✗ | ✔ | ✗ | ✗ | ✗ | ✔ | ✔ | ✗ | ✔ | ✗ |
| ClevrSkills [22] | ✗ | ✔ | ✗ | ✗ | ✗ | ✔ | ✔ | ✔ | ✔ | ✗ |
| LabUtopia (Ours) | ✔ | ✔ | ✔ | ✔ | ✔ | ✔ | ✔ | ✔ | ✔ | ✔ |

Table 1: Comparison with existing embodied AI simulators/benchmarks. *Fluid / Physics / Chemistry:* Support for simulating fluids, realistic physical interactions, and chemical processes, respectively. *Scene / Object:* Whether the benchmark supports realistic lab environments with scene-level and object-level assets. *Multi-action:* Multiple different axiom actions. *Composed:* Tasks involve the simple composition of atomic actions. *Generalization:* Generalization task across unseen scenes or object variations. *Long-Horizon:* Tasks demand high-level planning and long sequences of atomic and composed actions. *Navigation:* Tasks integrate spatial navigation with manipulations.

# 3   Laboratory Simulation Suite

We introduce **LabUtopia**, a high-fidelity simulation platform tailored to the challenges of embodied manipulation in laboratory settings. It is specifically designed for simulating, training, and evaluating agents in lab-centric tasks. LabUtopia consists of three key components: *LabSim* provides high-fidelity physical simulation with extensions for modeling chemically relevant dynamics, such as fluid mixing and reactive state transitions. *LabScene* includes a diverse asset library of scientific instruments and procedurally generates 3D environments, enabling rich spatial and task variations. Finally, a built-in trajectory collection module supports automated generation of expert demonstrations, facilitating scalable data collection for diverse lab tasks.

## 3.1   LabSim: High-Fidelity Simulation Environment

LabSim is a high-fidelity simulation engine designed to model the rich physical and chemical phenomena of laboratory environments. It not only supports physically accurate modeling of diverse material properties, but also introduces a reasoning-driven pipeline to simulate chemical reactions, enabling aligned training and evaluation with real-world scientific workflows.

**Physical Realism.** LabSim supports physically accurate interactions among rigid, deformable, and fluid entities [45]. Each asset in the environment is annotated with empirically grounded physical properties, we enable precise contact and collision modeling. For soft materials, we incorporate deformable body physics to capture compressible and elastic behavior. Notably, for fluid simulation, we employ a GPU-accelerated Position-Based Dynamics (PBD) framework [41], supporting rich fluid-agent interactions required for scientific manipulation.

**Chemical Process Modeling.** To simulate chemical processes within laboratory tasks, we introduce a chemical engine that integrates a curated knowledge base with a reasoning model. We begin by constructing a structured database of 200 common chemical substances, sourced from the authoritative PubChem repository [33]. Each encodes its key attributes, such as color, molar mass, and pH value, allowing it to be represented as a substance asset within the simulation. Given a set of reactants, we leverage a large language model (GPT-4o [28]) to reason about potential chemical processes and infer corresponding transformations, including color changes, product formation, etc. These inferred changes are then rendered in the simulation by dynamically updating the physical state and visual properties of the involved substances. This engine equips LabUtopia with the capability to model complex chemical interactions with both interpretability and flexibility.

## 3.2   LabScene: A Large-Scale Dataset of Scientific Laboratory Scenes and Instruments

Current 3D scene datasets primarily focus on domestic, office, or industrial environments [36, 57, 4, 17, 27], offering limited support for simulating laboratory settings. However, training and evaluating embodied agents in lab-centric tasks requires high-quality, interactive environments populated with

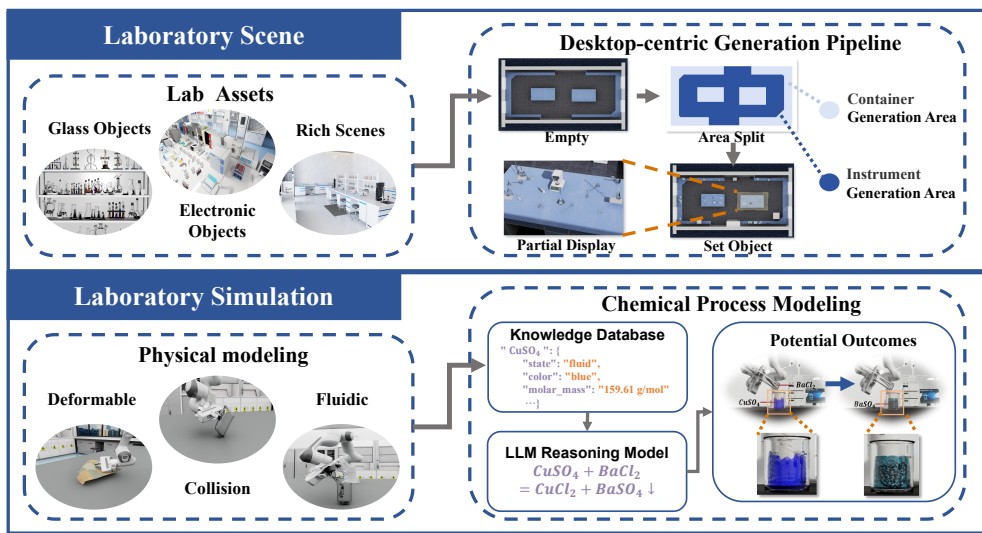

Figure 2: An overview of our laboratory simulation suite. **LabScene** automatically synthesizes scalable laboratory scenes using a diverse asset library and a procedural generation pipeline, while **LabSim** supports the simulation of high-fidelity physical and chemical interactions.

scientifically relevant instruments and layouts [38]. To address this gap, we introduce LabScene, a scalable dataset of laboratory object and scene assets with the procedural generation mechanism.

**Scene Assets.** Due to the scarcity of open-source lab assets in the community, we collected a large number of candidate scenes from designer websites. These raw assets underwent a multi-stage pre-processing pipeline, including content filtering, format normalization, and structural standardization. To ensure realism, we consulted experts in chemistry and physics to assess the fidelity of the scenes and provide refinement suggestions. Based on their feedback, we selected approximately 100 high-quality, expert-verified scenes to serve as the foundational environments for LabUtopia.

**Object Assets.** While some of the collected laboratory scenes already include basic instruments and furnishings, many lack the fine-grained internal structures and detailed geometry necessary for accurately simulating and executing real-world laboratory tasks. To overcome this limitation, we curated and constructed a comprehensive library of high-fidelity object assets that represent a wide range of laboratory tools and apparatus. These assets encompass essential equipment such as drying ovens, centrifuges, pipettes, and balances, as well as diverse glassware and plasticware types including beakers, flasks, test tubes, and Petri dishes. To ensure full compatibility with robotic manipulation and physics-based simulation, all assets were refined, standardized, and modularized into physically consistent, interactable forms. The final collection consists of approximately 60 categories of laboratory equipment and over 80 types of transparent glassware and plasticware, covering a rich diversity of materials, scales, and functional configurations.

**Environment Generation Pipeline.** To incorporate our collected instruments into the laboratory scenes, we develop an environment generation pipeline that balances layout diversity with physical plausibility. Specifically, all objects are placed sequentially based on importance and size rankings. For each object, candidate positions and orientations are sampled from a discretized grid that satisfies various constraints, including boundary, collision, and instrument-specific constraints [12, 60]. The layout score is computed considering factors such as edge proximity, inter-object distance, and orientation alignment. The configuration with the highest score is selected. If the random sampling fails to produce a valid layout within a time limit, the system falls back to a depth-first search strategy [55], systematically exploring placements while enforcing physical and spatial constraints. This hybrid approach ensures functional, reasonable scene layouts suitable for embodied agent training.

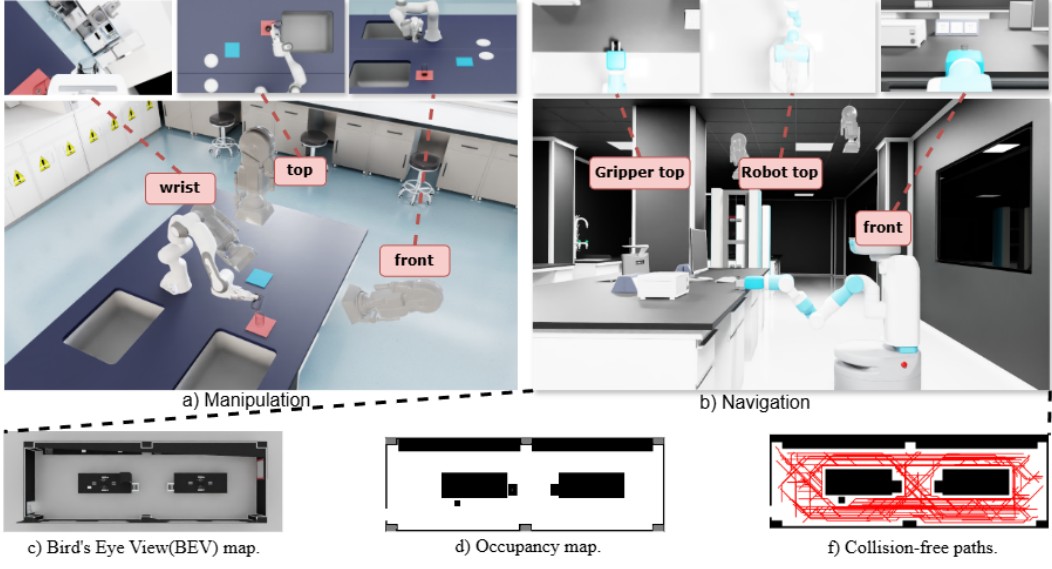

Figure 3: (a) Workspace for manipulation tasks. (b) Illustration of the navigation environment. (c) Bird's-eye view of the navigation map. (d) Occupancy grid map used for navigation. (f) Occupancy grid map with the planned path highlighted. '

### 3.3 Task-Rich Trajectory Collection

**Manipulation Trajectory Auto-collection.** We divide the motion planners into two levels: atomic action controllers and task-level action controllers. Atomic actions are standard laboratory operations, such as pouring and stirring, that align with experimental protocols and are executed using finite state machines. Task-level controllers organize atomic action controllers to collect data for specific tasks. For atomic actions, we control multiple target keypoints for the robotic arm. At each keypoint, an RMPflow controller [7] plans motion toward dynamically determined positions based on the real-time state of manipulated objects. We incorporate spherical linear interpolation (Slerp) [19] for continuously manipulating articulated objects, *e.g.,* opening the dry box task, for more robust results. Furthermore, task-level controllers organize atomic actions to streamline the entire experimental procedure, enabling our motion planner to generate demonstration data for imitation learning efficiently. During demonstration data collection, all objects are randomly initialized within a predefined spatial range to ensure diversity and generalization.

**Navigation Trajectory Auto-collection.** Robots in laboratory environments need to autonomously move between locations and interact with various experimental instruments. To automate trajectory collection, we design a method based on the A* algorithm [20] and occupancy map [14]. Specifically, we first generate and store occupancy maps built by Isaac Sim for the laboratory. These maps explicitly indicate the areas where obstacles are present, thus marking regions that are non-navigable. This information is then bound to the laboratory asset data. When deployed in a new laboratory scene, the robot uses its initial and target positions to plan a path with key waypoints. It then follows these waypoints, generating navigation trajectory data. This approach proves successful in experiments, offering an efficient solution for trajectory planning and data collection.

## 4 LabBench: Hierarchical Benchmark for Lab Agents

Embodied manipulation in laboratory environments spans a wide spectrum of tasks, ranging from low-level interactions to long-horizon workflows. These tasks vary significantly in complexity and skill requirements, making it difficult to evaluate agent capabilities in a unified manner. To address this, we propose LabBench, a hierarchical benchmark comprising over 50 tasks designed to systematically evaluate embodied agents across multiple levels of control, planning, and reasoning.

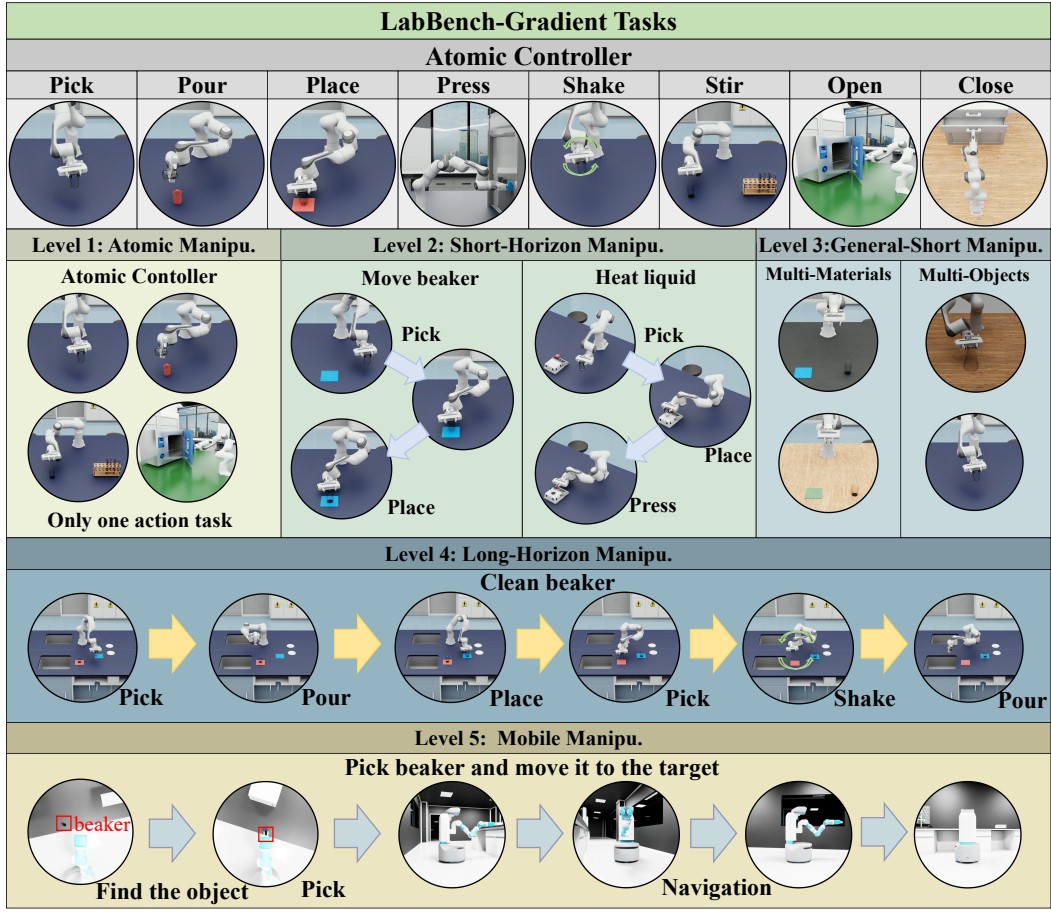

Figure 4: An overview of our hierarchical benchmark. **LabBench** structures scientific tasks across five levels, from atomic manipulations to long-horizon experiments, enabling rigorous evaluation of embodied agents in realistic laboratory settings.

## 4.1 Five-Level Task Structure

In order to comprehensively evaluate embodied agents in laboratory environments, LabBench organizes its tasks into five levels of increasing complexity, ranging from low-level atomic actions to integrated long-horizon and mobile manipulation tasks.

**Level 1: Atomic Manipulation Tasks.** This level focuses on fundamental low-level interactions that serve as the building blocks for more complex operations. Tasks include single-step action such as grasping, pouring, stirring, opening instrument, and placing containers. These actions can typically be executed via primitive controllers without requiring task-level planning.

**Level 2: Short-Horizon Manipulation Tasks.** This level involves agents performing a sequence of 2-3 atomic actions to complete a compound objective. For example, an agent might open a container and then pour a reagent, or pick up a test tube and place it in a mixer. These tasks require precise coordination of sequential actions to achieve the desired outcome.

**Level 3: Generalizable Short Manipulation Tasks.** This level evaluates agents' generalization capabilities under distributional shifts. Agents are trained jointly on mix of objects with varying shapes and appearances, as well as different visual material and environmental. Tasks are tested in novel scenarios featuring unseen object configurations, unfamiliar scene arrangements, or appearance variations. The evaluation tests the agents' capacity to transfer learned skills to effectively handle out-of-domain objects and scenes.

**Level 4: Long-Horizon Manipulation Tasks.** This level involves high-level planning and executing multi-step laboratory protocols that span numerous atomic and composed actions. These workflows, such as preparing a chemical solution or executing the cleaning instrument program, require high-level planning, reasoning, and robustness to compounding execution errors.

**Level 5: Mobile Manipulation Tasks.** The highest level integrates spatial navigation with manipulation. Agents are required to traverse large-scale laboratory environments using mobile-base control while performing manipulation tasks. Our task is to transport container between areas.

## 4.2 Evaluation Protocol

**Embodiment.** We employ a 7-DoF Franka Emika Panda [16] manipulator with a parallel gripper for manipulation tasks. For navigation tasks, we utilize the Fetch mobile manipulator [15] and a mobile manipulation robot composed of the Clearpath Robotics Ridgeback base integrated with a Franka Emika Panda arm [9], both supporting arm manipulation and base locomotion. These robots are controlled using three degrees of freedom—x and y velocities and rotation—enabling integrated navigation and manipulation [59].

**Evaluation Execution.** Object positions are randomly placed within a 15 cm × 15 cm region, adjusted according to their sizes and workspace constraints to ensure compatibility with task requirements. A task instance is considered successful if the current state remains within the tolerance threshold of the goal state and continuously satisfied for 2 seconds after completing the final task stage, consistent with prior works [19, 23, 26, 40]. For example, in the "Open Door" task, success requires the cabinet door to remain at the specified open position for 2 seconds after the robot releases the handle, ensuring no shortcuts during motion. The success rate is used as the evaluation metric in LabBench, with strict evaluation ensuring both the robot and object maintain the successful state for the required duration, only after the motion planner has executed its final action. More details about the task descriptions, visualizations, and success determination criteria can be found in Appendix C.

# 5 Experiment

## 5.1 Experimental Setup

**Models.** To benchmark the performance of existing imitation learning algorithms in LabSim, we select two representative models: ACT [18], Diffusion Policy [8], and $\pi_0$ [6]

• ACT is a transformer-based model designed for action chunk prediction in robotic manipulation. ACT processes RGB images, robot proprioception through a multi-layer transformer encoder. At each decision step, ACT autoregressively predicts the next low-level action, conditioned on past observations and actions.

• Diffusion Policy is a generative model that formulates robot control as a conditional diffusion process. The model takes recent observations as input and learns to generate control trajectories by progressively denoising an initial random trajectory sample, conditioned on the observation context. In our work, we utilize the CNN-based version of Diffusion Policy.

• $\pi_0$ is a vision-language-action (VLA) model based on PaliGemma, which enables modeling of complex continuous actions by incorporating an additional action expert output module and utilizing flow matching techniques.

Table 2: Performance comparison across task levels. Values represent success rates (%).

| Task Level | Task Name | ACT | DP |
|---|---|---|---|
| Level-1 | Stir | 86.7 | 95.0 |
| | Pick | 75.0 | 86.7 |
| | Pour | 76.7 | 73.3 |
| | Press | 93.3 | 96.7 |
| | Place | 73.3 | 80.0 |
| | Shake | 93.3 | 86.7 |
| | Open Door | 77.5 | 67.5 |
| | Close Door | 71.7 | 65.0 |
| | Open Drawer | 93.3 | 95.0 |
| | Close Drawer | 96.7 | 100.0 |
| Level-2 | Pour Liquid | 67.5 | 50.0 |
| | Shake Beaker | 77.5 | 67.5 |
| | Heater Beaker | 86.7 | 25.0 |
| | Operate Drawer | 73.3 | 10.0 |
| | Stir w/ GlassRod | 55.0 | 10.0 |
| | Transport Beaker | 78.3 | 50.0 |

Table 3: Performance comparison across Level-3 task. **Results are reported under ID/OOD settings.** Values represent success rates (%).

| Task Level | Task Name | $\pi_0$ | ACT | DP |
|---|---|---|---|---|
| Level-3 | Pick | 83.3/85.8 | 81.7 / 71.7 | 53.3 / 41.7 |
| | Press | 92.5/89.1 | 98.3 / 96.7 | 81.7 / 31.6 |
| | Open Door | 51.6/53.3 | 73.3 / 65.0 | 63.3 / 58.3 |
| | Pour Liquid | 40.0/38.3 | 75.0 / 65.0 | 46.6 / 31.6 |
| | Heater Beaker | 89.1/86.7 | 86.7 / 80.0 | 21.6 / 8.3 |
| | Transport Beaker | 86.7/88.3 | 77.5 / 73.3 | 67.5 / 15.0 |

**Model Input And Output.**

All manipulation tasks utilize three cameras (a wrist camera on the robotic arm, one facing the robotic arm, and one directly overhead), with each camera outputting 256×256 RGB images by default. DP and ACT models use images from the overhead and front-facing cameras, while $pi_0$ model uses images from the wrist camera and the camera facing the robotic arm. Users can also render images at any resolution or output depth maps. Other Omniverse sensors (e.g., tactile sensors) are currently disabled as they are unnecessary for the current tasks, but full support is available if needed. The input for all models also includes the absolute values of the robot's joint positions. The model's output is the predicted joint positions of the robotic arm for the next time step.

**Training Details** Our training parameters are primarily adapted from the open-source ACT, Diffusion Policy and OpenPI algorithms. ACT and Diffusion Policy were implemented using PyTorch, with training and evaluation conducted on a single NVIDIA RTX 4090 GPU. $\pi_0$ was implemented using JAX, with training on 8 A800 80G GPUs. ACT and Diffusion Policy were trained for 200 epochs using the AdamW optimizer and Exponential Moving Average (EMA). $\pi_0$ was trained for 30,000 steps. Model inputs include two camera views, each with a resolution of $256 \times 256$ pixels, as well as robot joint position inputs. The output consists of predicted joint parameters. A fixed learning rate of $1 \times 10^{-4}$ was used throughout. The batch sizes were set to 128 for ACT and 64 for Diffusion Policy. Additionally, ACT and $\pi_0$ were trained using a single frame of historical observation, whereas Diffusion Policy was trained using three consecutive frames as input. The action chunk lengths for ACT, DP, and $\pi_0$ are 60, 60, and 8, respectively.

## 5.2 Experimental Results

We evaluated the performance of the ACT and DP models across Level-1 to Level-3 tasks in Table 2 and Table 3, and additionally tested the $\pi_0$ model on Level-3 tasks. For Level-1 tasks, both models demonstrated robust fundamental manipulation abilities, achieving high success rates due to the simplicity and single-step nature of these tasks. In Level-2 tasks, the success rates began to decline, particularly for the DP model. We observed that DP frequently stalled during execution. For example, in the Heater Beaker task, the DP algorithm paused during the final action until the time limit expired; while in the Operate Drawer task, most DP failures resulted from its inability to successfully execute any action. In the Stir with GlassRod task, both models exhibited noticeable errors in grasping or stirring positions due to the small size of the glass rod.

In Level-3 tasks, the performance gap widened, with ACT maintaining superior stability, while DP struggled with generalization to novel materials and precise actions, such as pressing a button in Heater Beaker. In contrast, ACT showed greater robustness when handling out-of-domain visual features. Additionally, we evaluated the VLA-based model $\pi_0$ on selected Level-3 generalization tasks. While pretrained VLA models, after fine-tuning, exhibited minimal performance degradation on out-of-distribution visual inputs or novel materials, they did not consistently outperform models trained from scratch in these tasks. We tested generalization to out-of-domain shapes in Table 4 by jointly training datasets with objects of varying sizes and evaluating

Table 4: Performance comparison of different models when manipulating different size objects. Values represent success rates (%).

| Task Level | Model | Success Rate (%) | |
|---|---|---|---|
| | | ID | OOD |
| Pick | ACT | 31.2 | 1.7 |
| | DP | 11.2 | 0.0 |
| Pour Liquid | ACT | 25.6 | 0.0 |
| | DP | 8.0 | 0.0 |

Table 5: Performance comparison of different models on Level-4 long-horizon tasks. SP: Single-stage Primitive; A1–A7: Different sub-steps of task sequence.

| Level-4 Task | Model | Success Rate (%) | | | | | | | |
| --- | --- | --- | --- | --- | --- | --- | --- | --- | --- |
| | | SP | A1 | A2 | A3 | A4 | A5 | A6 | A7 |
| Clean Beaker | ACT | 14.0 | 99.3 | 51.9 | 43.3 | 42.5 | 12.3 | 10.6 | 1.6 |
| | DP | 2.5 | 92.7 | 8.1 | 1.3 | 0.0 | 0.0 | 0.0 | 0.0 |
| Drying Beakers | ACT | 6.3 | 81.3 | 42.0 | 19.7 | 9.3 | 3.5 | 1.2 | 0.0 |
| | DP | 2.0 | 76.0 | 32.0 | 26.3 | 0.0 | 0.0 | 0.0 | 0.0 |
| Liquid Fusion | ACT | 8.3 | 96.3 | 57.6 | 46.7 | 42.5 | 12.3 | 10.6 | 1.6 |
| | DP | 6.7 | 93.5 | 12.3 | 8.3 | 0.0 | 0.0 | 0.0 | 0.0 |

on out-of-domain objects. The results revealed that joint training with objects of significantly different sizes led to a substantial drop in in-domain success rates, with near-zero success rates for out-of-domain shapes and sizes. This suggests that both models largely lack the ability to manipulate out-of-domain objects.

In Level-4 tasks, as shown in Table 5, we assessed the performance in long-horizon tasks. ACT significantly outperformed DP, achieving higher success rates in single-stage primitive (SP) actions and sub-steps (A1–A7), though both models experienced sharp declines in later sub-steps (A5–A7) due to cumulative errors in complex sequences. In sub-steps, although each policy is more stable when handling its own task, it is necessary to design specific action decomposition and switching mechanisms, which increases the overall system complexity. Moreover, the transitions between actions may become discontinuous or suffer from distribution shift issues.

ACT exhibited greater stability across most subsequent tasks. We hypothesize that this disparity arises from several factors. First, compared to ACT, DP's shorter prediction horizon makes it more prone to stagnation. For example, in the Heater Beaker task, DP often hovered above the button without pressing it in the final step. Our task setting outputs joint position, and DP's outputs are more jittery than ACT's, increasing the likelihood of dropping grasped objects. Second, DP's high sensitivity to visual features led to inconsistent performance, particularly in the Stir with GlassRod task, where its output struggled to accurately locate the glass rod. Furthermore, in generalization tests for Level-3 tasks, DP experienced significant performance declines when encountering unseen materials. These findings suggest that future work should focus on enhancing the models' ability to generalize across diverse object properties and task variations. Please refer to the Supplementary Materials for more details.

## 6 Conclusion and Limitations

We propose a simulation and evaluation platform for embodied agents in scientific laboratory settings, designed to provide researchers with a unified environment for training, testing, and analyzing agent performance in complex experimental tasks. To this end, we have designed a hierarchical benchmark covering five levels and more than 30 tasks, and constructed a high-fidelity simulation environment for physical and chemical processes, along with a rich library of laboratory assets. We conducted a systematic evaluation of mainstream imitation learning methods, and the results show that current imitation learning approaches still exhibit significant shortcomings when tackling complex long-horizon tasks. While these methods demonstrate some degree of visual generalization, they show little to no generalization to objects with different shapes. Overall, these findings indicate that, despite notable progress in embodied agents and their underlying models, achieving general-purpose agents for scientific laboratory scenarios will require continued in-depth research and innovation. Although LabUtopia provides a scalable and versatile simulation platform that supports tasks across different levels and types, several limitations remain. First, the current benchmark operates entirely within simulation. Bridging the sim-to-real gap by constructing a physical laboratory environment represents an important direction for future work. Second, LabUtopia currently supports only two robot embodiments (Fetch and Panda); expanding support to additional robot types would facilitate the collection of more diverse action trajectories and further promote research on generalization.

## Acknowledgements

This work was supported by the Shanghai Municipal Science and Technology Major Project. This work was supported by the JC STEM Lab of AI for Science and Engineering, funded by The Hong Kong Jockey Club Charities Trust, the MTR Research Funding (MRF) Scheme (CHU-24003), the Research Grants Council of Hong Kong (Project No. CUHK14213224).

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

# ————Appendix————

The structure of Appendix is as follows:

- Appendix A More Details of LabScene.

- Appendix B More Details of LabSim.

- Appendix C More Details of LabBench.

- Appendix D Impact Statement.

## A  More Details of LabScene

### A.1  Asset in OpenUSD Format

OpenUSD (Universal Scene Description), developed by Pixar and integrated into NVIDIA's Omniverse platform, is an open-source 3D scene description framework and file format designed for efficient interchange and collaboration across diverse 3D graphics workflows. It uses a hierarchical structure to represent complex scenes, with core components like Stages, Prims, Attributes, and Relationships. OpenUSD supports extensible schemas, real-time rendering, and non-destructive editing, enabling seamless asset sharing and simultaneous collaboration across tools like Blender, Unreal Engine. In Omniverse, OpenUSD is enhanced with NVIDIA RTX rendering and APIs, facilitating applications in film, gaming, architecture, robotics, and digital twins, with benefits like reduced data duplication and flexible pipeline design.

#### A.1.1  Container Assets

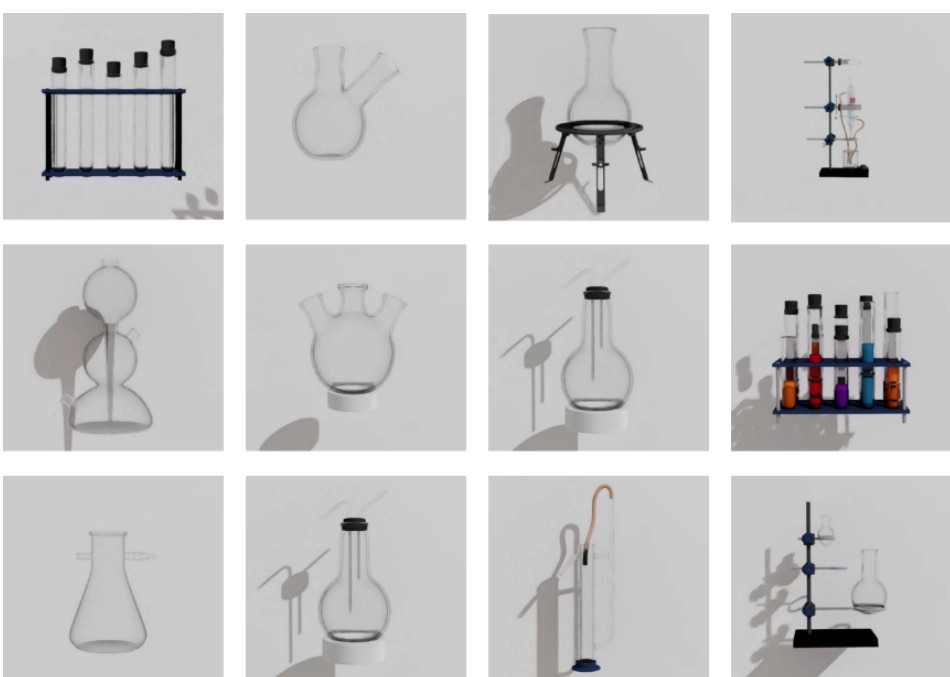

### A.1.2 Instrument Assets

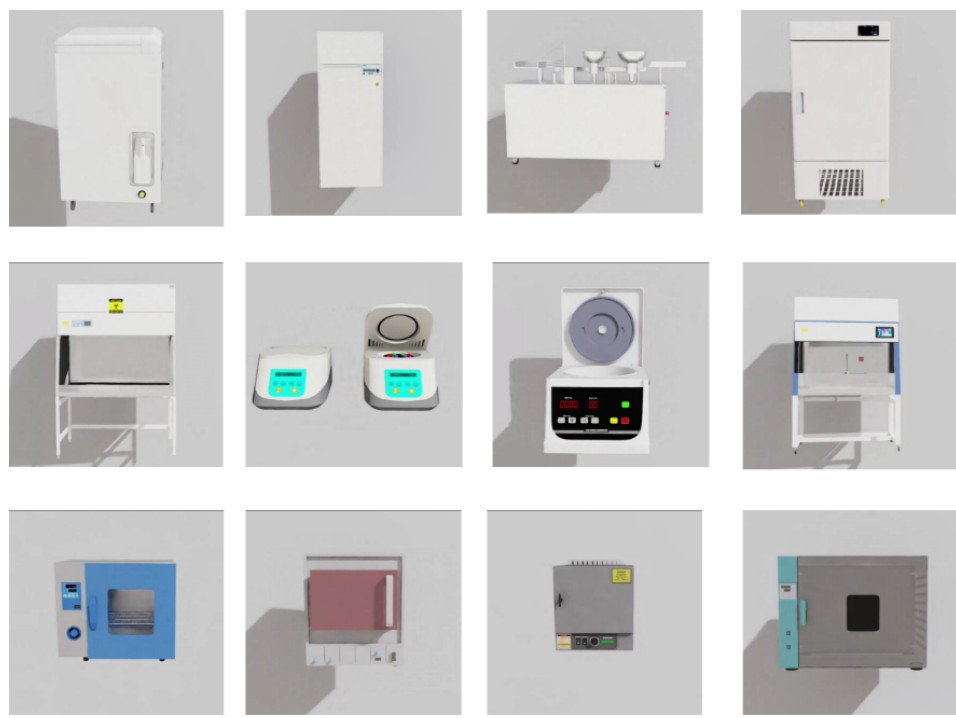

### A.1.3 Lab Scene Assets

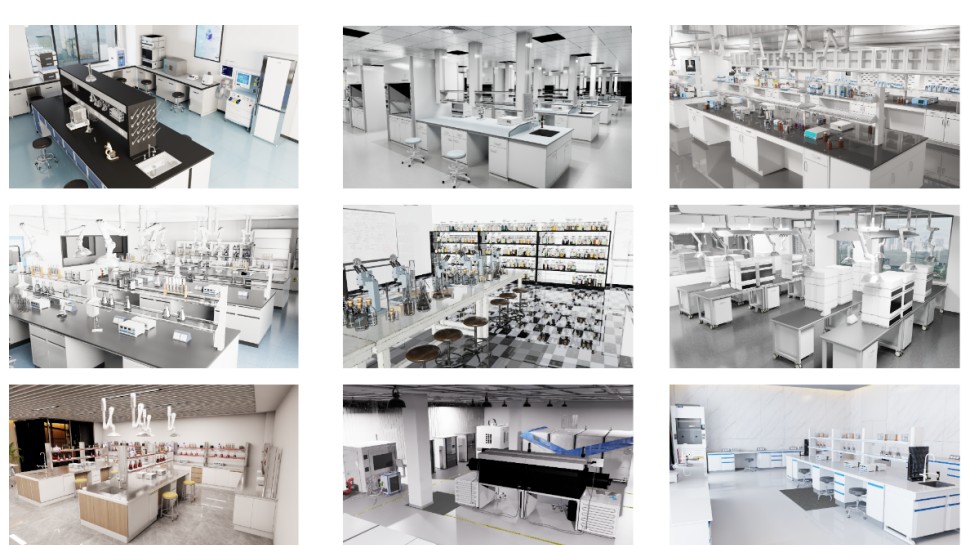

### A.2 Speed

Using two 256×256 cameras, our system runs at 23 fps for liquid simulation and 45 fps for rigid body simulations on an NVIDIA RTX 4090 GPU and Intel(R) Xeon(R) w5-2445. If cameras are not used, the simulation runs at approximately 115 fps.

# B  More Details of LabSim

## B.1  Database of Chemical Substances

To enable realistic chemical process simulation, we constructed a structured chemical knowledge base comprising 200 representative inorganic and organic compounds. All data were collected from authoritative chemical repositories (e.g., PubChem) to ensure the consistency and reliability of physicochemical information. For each compound, the following key properties were extracted and normalized: physical attributes (color, appearance, density, melting point, boiling point), chemical attributes (molecular formula, molar mass, solubility), and optional safety or reactivity descriptors. All records were serialized into a JSON-based schema for efficient retrieval and integration within the simulation engine. An example subset of this data structure is shown below.

Listing 1: Example JSON entries from the structured chemical knowledge base

```
1  [
2      ...
3    {
4      "compound_name": "Aluminum oxide",
5      "chemical_formula": "Al2O3",
6      "molar_mass": "101.9612772 g/mol",
7      "density": "3.9870 g/cm3",
8      "melting_point": "2,072.00 °C",
9      "boiling_point": "2,977.00 °C",
10     "solubility": "insoluble",
11     "sublimation_point": null,
12     "appearance": "white solid"
13   },
14   {
15     "compound_name": "Silicon dioxide",
16     "chemical_formula": "SiO2",
17     "molar_mass": "60.0843 g/mol",
18     "density": "2.6480 g/cm3",
19     "melting_point": "1,713.00 °C",
20     "boiling_point": "2,950.00 °C",
21     "solubility": null,
22     "sublimation_point": null,
23     "appearance": "Transparent or white"
24   },
25     ...
26 ]
```

## B.2  Process Reasoning Pipeline

The chemical process reasoning pipeline translates a set of input reactants and experimental conditions into plausible chemical transformations that can be enacted by the simulation engine. The pipeline couples (i) deterministic knowledge retrieval from the structured chemical knowledge base with (ii) generative inference performed by a large language model. The following describes the designed pipeline stages, the inter-stage data contracts, and example prompts/tool-calls to drive the reasoning flow.

**1. Input parsing and normalization.** The pipeline accepts an input specification containing reactant identities (names or database IDs), amounts (optional), and contextual parameters (temperature, solvent, concentration, catalysts, pH). Input names are normalized via a canonicalization step that maps user strings to knowledge-base compound identifiers.

**2. Knowledge retrieval (tool-like call).** Relevant physicochemical fields are retrieved from the JSON knowledge base using a deterministic query. Example tool-like invocation:

`

Listing 2: Example chemical-database query

```
1  CALL chem_db.lookup({
2    "compounds": ["Hydrochloric acid", "Sodium hydroxide"],
3    "fields": [
4      "compound_name",
5      "chemical_formula",
6      "molar_mass",
7      "density",
8      "melting_point",
9      "boiling_point",
10     "solubility",
11     "sublimation_point",
12     "appearance"
13   ]
14 })
15 -- returns: JSON array of compound records as in Appendix B.1
```

The retrieved records are attached to the reasoning context to supply the LLM with grounded, factual attributes.

**3. Reaction inference (LLM-assisted).** A structured prompt is passed to the LLM. The prompt uses a two-part format: a short system instruction (model role + safety constraints) and a user instruction with the normalized reactant records and explicit task. The model is requested to respond in machine-readable JSON only, conforming to the specified output schema.

Listing 3: LLM prompt template for reaction inference

```
1  SYSTEM:
2  You are a chemical process reasoning assistant. Use only the provided
       compound attributes to infer plausible qualitative reactions.
3  Follow safety rules:
4  (1) do not propose energetic detonation sequences,
5  (2) avoid suggesting synthesis of controlled or hazardous compounds,
6  (3) prefer qualitative descriptions that are experimentally
       interpretable.
7  Return results as JSON conforming exactly to the requested schema.
8
9  USER:
10 Context:
11 - Reactants: ["Hydrochloric acid", "Sodium hydroxide"]
12 - Experimental conditions: {"solvent":"water", "temperature":"25 C", "
       concentration": "0.1 M", "open_system": true}
13 - Retrieved compound records: <attach JSON records from chem_db.lookup
       >
14
15 Task:
16 1. Infer plausible reaction(s) between the reactants under the given
       conditions.
17 2. For each inferred reaction, provide:
18    - reaction_id
19    - reaction_type (e.g., neutralization, precipitation, redox,
          hydrolysis)
20    - stoichiometry (if obvious qualitatively)
21    - expected_products (list of product objects with names and basic
          attributes)
22    - observable_changes (color change, gas evolution, precipitation,
          temperature change)
23    - confidence_score (0.0 - 1.0)
24 3. Provide a short rationale (1-2 sentences).
25
26 Return ONLY JSON following this schema:
27 {
```

```
28    "reactions": [
29      {
30        "reaction_id": "r1",
31        "reaction_type": "...",
32        "stoichiometry": "...",
33        "expected_products": [
34          {"compound_name":"...", "chemical_formula":"...", "phase":"aq
                /solid/gas", "qualitative_yield":"high/medium/low or null
                "}
35        ],
36        "observable_changes": ["..."],
37        "confidence_score": 0.0,
38        "rationale":"..."
39      }
40    ]
41 }
```

# C  More Details of LabBench

## C.1  Task Details

### C.1.1  Level 1:Atomic Manipulation Tasks:

1. **Pick:**

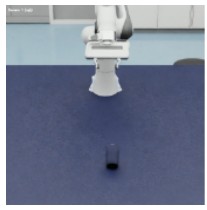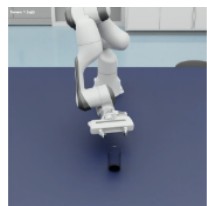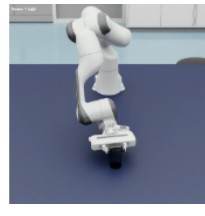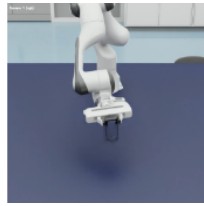

- **Description:** The atomic action "Pick" is designed to grasp a chemical container (e.g., a beaker or conical flask) from a desktop, managing the retrieval process through multiple phases: initially (Phase 0), the end effector is positioned above the object; then (Phase 1), it is lowered closer to the object; (Phase 2) the end effector is aligned for grasping; (Phase 3) a pause allows the robot's dynamics to stabilize; (Phase 4) the gripper closes to secure the object; (Phase 5) the object is lifted; and finally (Phase 6), the sequence is completed.
- **Success Criteria:** The object must be securely grasped without being knocked over. Following grasping, it must be lifted to a height exceeding 20 cm and maintained steadily for a designated period, with success defined as the beaker remaining intact and not falling.

2. **Pour:**

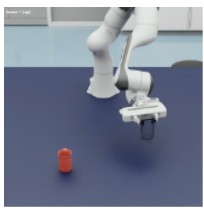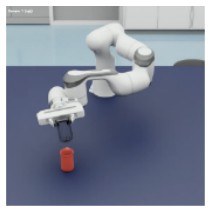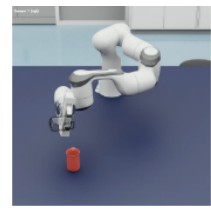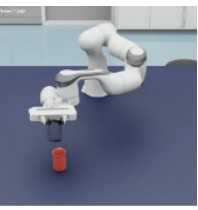

- **Description:** The atomic action "Pour" is designed to pour liquid from a previously grasped chemical container (e.g., a beaker or conical flask) into a designated target container or location and subsequently return to an upright position, managing the process through multiple phases: initially (Phase 0), the end effector is positioned at the pouring start location; then (Phase 1), it adjusts to a ready position for grasping or pouring; (Phase 2) the pouring begins with the container tilted; (Phase 3) a pause maintains the container in a stationary state; (Phase 4) the container is returned to an

upright position through reverse tilting; and finally (Phase 5), the pouring sequence is completed with the container stabilized.

- **Success Criteria:** As this action relies on "Pick" as a prerequisite, the task design focuses solely on the "Pour" phase, recording data for this segment and evaluating success rates only after a successful "Pick." Success is defined by the container being tilted and poured within a specified region above the target, achieving a tilt angle greater than a predetermined threshold during pouring, returning to an upright position with a tilt angle less than a specified threshold, and remaining stable for a designated period.

3. **Place:**

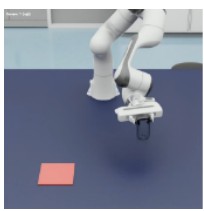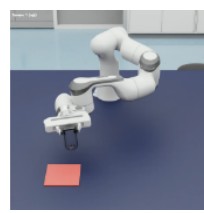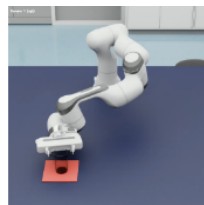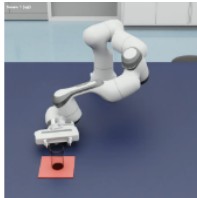

- **Description:** The "Place" atomic action enables a robotic arm to set a chemical container (e.g., a beaker or conical flask) from its gripper onto a designated lab bench surface, executing the placement process through multiple phases: initially (Phase 0), the end effector positions the container above the target area; then (Phase 1), it lowers the container closer to the surface; (Phase 2), the container's base is aligned for placement; (Phase 3), a pause stabilizes the robotic arm's dynamics; (Phase 4), the gripper opens smoothly to release the container; (Phase 5), the end effector retracts above the surface; and finally (Phase 6), the sequence is completed.

- **Success Criteria:** As the "Place" action follows a successful "Pick," the task evaluation focuses solely on the placement phase. Success is defined by the container being placed at the target location on the lab bench, remaining upright and undamaged with its base in full contact with the surface, without falling, tipping or colliding with other objects, and maintaining stability for a designated period after placement.

4. **Press:**

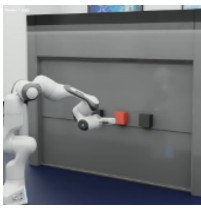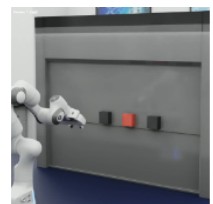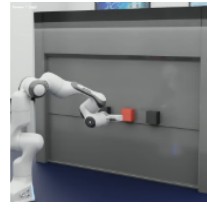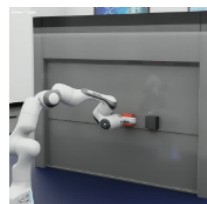

- **Description:** The "Press" atomic action enables a robotic arm to press a button on a laboratory instrument, executing the pressing process through multiple phases: initially (Phase 0), the end effector moves to a position above the target button with an initial offset; then (Phase 1), the gripper adjusts to a specified spacing to prepare for pressing; (Phase 2), the end effector executes the pressing action to activate the button; and finally (Phase 3), the sequence is completed.

- **Success Criteria:** The end effector is precisely aligned with the target button, presses the button a sufficient distance to activate it, and maintains stable contact for a designated period, with success confirmed by the button's movement without misalignment.

5. **Shake:**

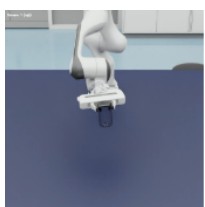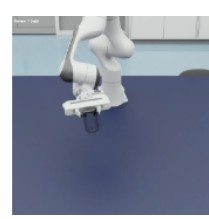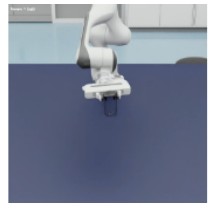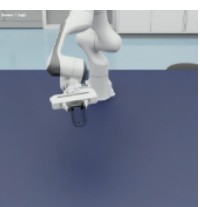

- **Description:** The "Shake" atomic action enables a robotic arm to agitate a chemical container (e.g., a beaker or conical flask) held by its gripper to mix its contents, executing the shaking process through multiple phases: initially (Phase 0), the end effector moves the container to an initial position above the workspace; then (Phase 1), it pauses to stabilize the initial position; (Phase 2), the container is tilted to the left for the first shake; (Phase 3), it is tilted to the right for the first counter-shake, with this left-right shaking sequence repeated three times; (Phase 8), the end effector returns the container to the initial position; (Phase 9), it pauses to stabilize the initial position; and finally (Phase 10), the sequence is completed.
- **Success Criteria:** As the "Shake" action follows a successful "Pick," the task evaluation focuses solely on the shaking phase. Success is defined by the chemical container completing three full left-right shake cycles, returning to the initial position, and maintaining stability for a designated period without spilling, tipping, or colliding with other objects.

6. **Stir**

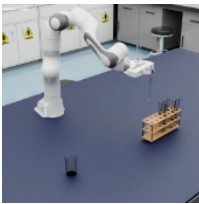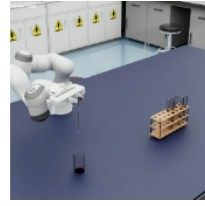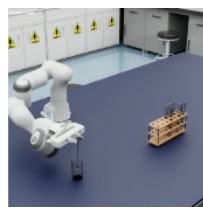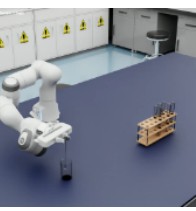

- **Description:** The "Stir" atomic action enables a robotic arm to manipulate a glass rod for stirring the contents of a target container (e.g., a beaker) held in place on the workspace. The stirring process proceeds through multiple structured phases: initially (Phase 0), the end effector lifts the glass rod slightly above the gripping position; (Phase 1), it moves the rod horizontally to a position directly above the target beaker; (Phase 2), the rod is inserted vertically into the beaker to a predefined depth; (Phase 3), the end effector executes a circular stirring motion within the container, typically following a clockwise or counterclockwise path with controlled speed and radius; (Phase 4), the rod is retracted vertically out of the beaker; and finally (Phase 5), the action completes with the end effector holding the rod above the beaker, ready for the next operation or placement.
- **Success Criteria:** As the "Stir" action follows a successful "Pick," the task evaluation focuses solely on the stirring phase. Success is defined by the glass rod being inserted into the target beaker region without interruption or hesitation, executing a continuous stirring motion, and upon completion, being lifted upward and fully withdrawn from the beaker area without causing spillage, collision, or deviation from the defined stirring trajectory.

7. **Open Door**

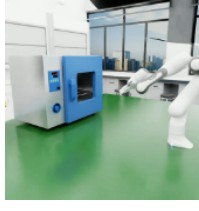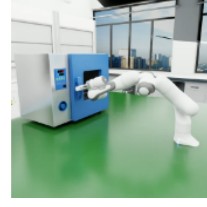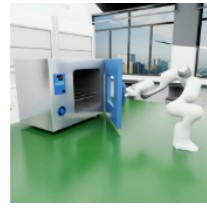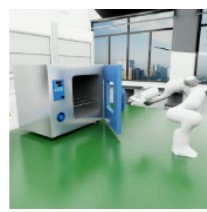

- **Description:** The "Open" atomic action enables a robotic arm to open a chemical instrument cabinet door or similar enclosure, simulating a human-like opening motion. The process is divided into several structured phases: initially (Phase 0), the end effector moves toward the target object, positioning itself near the handle or designated opening area; (Phase 1), the gripper performs a rotational motion (e.g., clockwise or counterclockwise) to simulate turning a handle or triggering a latch mechanism; (Phase 2), the gripper applies a pulling force to move the object (e.g., door or drawer) along a predefined direction, initiating the opening; (Phase 3), the system evaluates whether the door or drawer has reached the designated open position. If the target position is

not reached, the pulling continues incrementally; (Phase 4), once the opening motion is complete, the arm maintains a stable posture for a short pause to ensure the object remains in the open state and vibrations subside; finally (Phase 5), the gripper releases the object, marking the completion of the entire opening process.

- **Success Criteria:** The robotic arm must accurately grasp the cabinet door handle without slipping or displacing it. Following grasping, the arm must rotate the handle along the door's hinge axis, opening the cabinet door to a predefined target angle. After reaching the target angle, the gripper releases the door handle and moves away. Success is defined by the door remaining open at the target angle without unintended movement or closing.

8. **Close Draw**

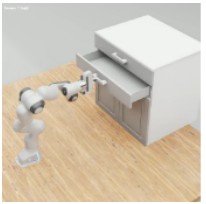 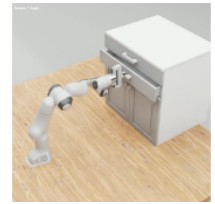 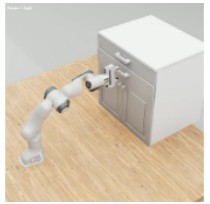 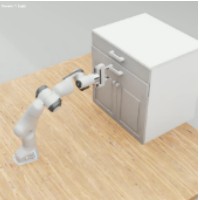

- **Description:** The "Close" atomic action enables a robotic arm to close a chemical instrument cabinet door or similar enclosure, replicating a natural closing motion. The process proceeds through several phases: initially (Phase 0), the end effector moves toward the door handle, positioning itself to grasp the handle securely; (Phase 1), the gripper rotates together with the door handle along the hinge axis, moving the door toward the closed position; (Phase 2) and subsequent phases involve the action completion stage, where the system sends no further motion commands and maintains a stationary posture, ensuring the door remains closed and stable.

- **Success Criteria:** The robotic arm must accurately grasp the cabinet door without slipping or misalignment. Following grasping, the arm must push the door along its track or hinge axis smoothly until the door is fully closed. Success is defined by the door reaching the fully closed position securely, with no gaps or unintended openings, and the gripper releasing or holding position without causing door misalignment or damage.

### C.1.2 Level 2: Short-Horizon Manipulation Tasks.

1. **Pour Liquid**

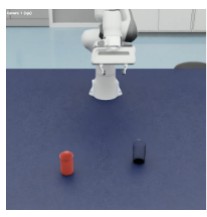 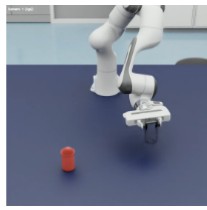 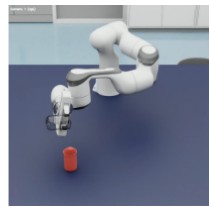 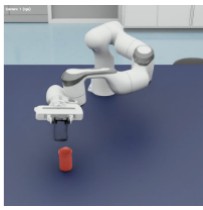

- **Description:** The Pour Beaker task is a medium-difficulty robotic operation that integrates the atomic actions "Pick" and "Pour" to securely grasp a chemical container (e.g., a beaker or conical flask) from a desktop and transport it to a designated location for pouring, while ensuring the container does not drop. The process begins with the Pick action, where the robot's end effector grasps the container through a precise sequence of positioning, lowering, aligning, stabilizing, gripping, and lifting. This is followed by the Pour action, where the end effector transports the container to the target location, aligns it for pouring, tilts it to dispense the contents, stabilizes, returns the container to an upright position, and completes the sequence, maintaining stability throughout to prevent dropping or spillage.

- **Success Criteria:** The chemical container (e.g., a beaker or conical flask) must be securely grasped without being knocked over during the Pick action. Following grasping, the container must be lifted to a height exceeding 20 cm, transported steadily to the designated target location without dropping, and successfully tilted to pour the

contents at the target position during the Pour action. After pouring, the container must be returned to an upright state and released without falling, ensuring the beaker remains intact and stable throughout the entire process.

2. **Shake Beaker**

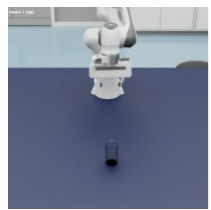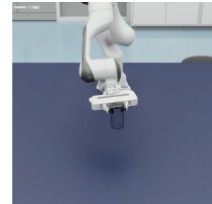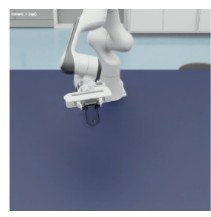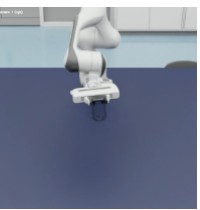

- **Description:** The Shake Beaker task is a medium-difficulty robotic operation that utilizes the atomic action "Pick" to securely grasp a chemical container (e.g., a beaker or conical flask) from a desktop, perform three side-to-side shakes, and stabilize it at the initial position, while ensuring the container does not drop. The process involves the Pick action, where the robot's end effector grasps the container through a precise sequence of positioning, lowering, aligning, stabilizing, gripping, and lifting. Subsequently, the end effector executes three controlled side-to-side shakes, returns the container to the initial position, and stabilizes it for a designated period, maintaining stability throughout to prevent dropping or spillage.

- **Success Criteria:** The chemical container (e.g., a beaker or conical flask) must be securely grasped without being knocked over during the Pick action. Following grasping, the container must be lifted to a height exceeding 20 cm, shaken side-to-side at least three times without dropping, and returned to the initial position. The container must then be stabilized for a designated period, ensuring it remains intact and stable throughout the entire process without falling or spilling.

3. **Transport Beaker**

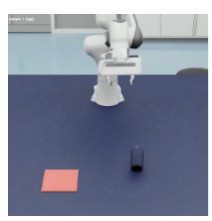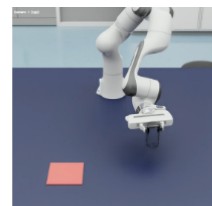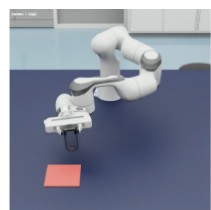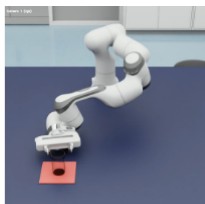

- **Description:** The Transport Beaker task is a medium-difficulty robotic operation that integrates the atomic actions "Pick" and "Place" to securely grasp a chemical container (e.g., a beaker or conical flask) from a desktop, transport it to a designated location, and place it down, while ensuring the container does not drop. The process begins with the Pick action, where the robot's end effector grasps the container through a precise sequence of positioning, lowering, aligning, stabilizing, gripping, and lifting. This is followed by the Place action, where the end effector transports the container to the specified location, lowers it to place it within a designated range, and releases it, ensuring the beaker remains upright and stable at the target position without falling or spilling throughout the process.

- **Success Criteria:** The chemical container (e.g., a beaker or conical flask) must be securely grasped without being knocked over during the Pick action. Following grasping, the container must be lifted to a height exceeding 20 cm and transported to the designated location without dropping. The container must then be placed down within the specified target area and remain upright, ensuring it remains intact and stable throughout the entire process without falling or spilling.

4. **Heater Beaker**

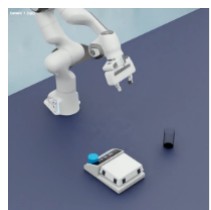 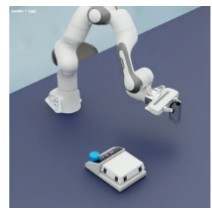 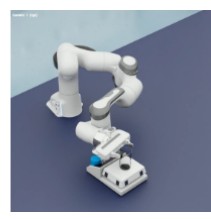 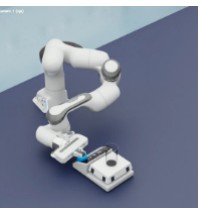

- **Description:** The Heater Beaker task is a medium-difficulty robotic operation that integrates the atomic actions "Pick," "Place," and "Press" to securely grasp a chemical container (e.g., a beaker or conical flask) containing a solution from a desktop, transport it to a heating device, place it on the device, and activate the heating process, while ensuring the container does not drop. The process begins with the Pick action, where the robot's end effector grasps the container through a precise sequence of positioning, lowering, aligning, stabilizing, gripping, and lifting. This is followed by the Place action, where the end effector transports the container to the heating device, lowers it to position it within the designated range of the device, and releases it, ensuring the beaker remains upright and stable. Finally, the Press action involves the end effector moving to the heating device's button, pressing it with sufficient force over a specified distance to activate the heating process, maintaining stability throughout to prevent the container from falling or spilling.

- **Success Criteria:** The chemical container (e.g., a beaker or conical flask) must be securely grasped without being knocked over during the Pick action. Following grasping, the container must be lifted to a height exceeding 20 cm and transported to the heating device without dropping. The container must then be placed down within the specified range of the heating device and remain upright. Finally, the heating button must be pressed with sufficient force over a specified distance to activate the heating process, ensuring the container remains intact and stable throughout the entire process without falling or spilling.

5. **Stir GlassRod**

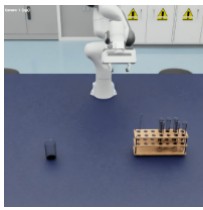 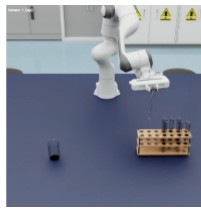 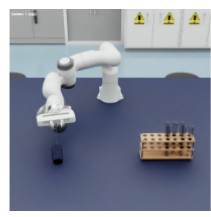 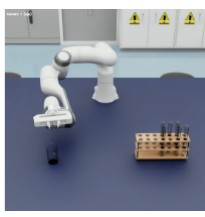

- **Description:** The Stir GlassRod task is a medium-difficulty robotic operation that integrates the atomic actions "Pick" and "Stir" to securely grasp a glass rod from a desktop and use it to stir a solution within a chemical container (e.g., a beaker or conical flask), while ensuring the rod does not drop. The process begins with the Pick action, where the robot's end effector grasps the glass rod through a precise sequence of positioning, lowering, aligning, stabilizing, gripping, and lifting. This is followed by the Stir action, where the end effector moves the glass rod to the chemical container, positions it within the solution, and performs a controlled stirring motion, maintaining stability throughout to prevent the rod from slipping or causing spillage of the solution.

- **Success Criteria:** The glass rod must be securely grasped without being knocked over during the Pick action. Following grasping, the rod must be lifted to a height exceeding 20 cm and accurately positioned within the chemical container (e.g., a beaker or conical flask) without causing additional collisions. The rod must then perform a controlled stirring motion within the solution without slipping or causing spillage. After stirring, the glass rod must be lifted from the container and stabilized for a designated period, ensuring it remains intact and stable throughout the entire process without dropping or causing unintended collisions.

6. **Operate Drawer**

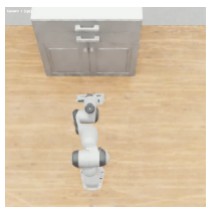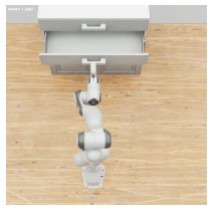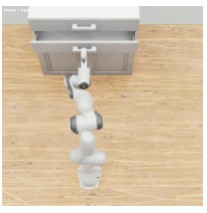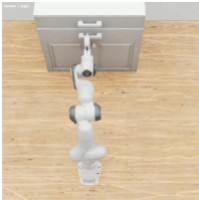

- **Description:** The Operate Drawer task is a medium-difficulty robotic operation that integrates the atomic actions "Open" and "Close" to smoothly operate a chemical container storage drawer, ensuring the drawer is opened and closed without disruption. The process begins with the Open action, where the robot's end effector approaches the drawer's handle through a precise sequence of positioning, aligning, stabilizing, gripping, and pulling to open the drawer smoothly. This is followed by the Close action, where the end effector repositions to the handle, grips it, and pushes the drawer to close it, maintaining controlled motion throughout to prevent the robotic arm from causing additional collisions with the drawer or surrounding objects.

- **Success Criteria:** The drawer's handle must be securely grasped without being knocked or misaligned during the Open action. Following grasping, the handle must be pulled smoothly to open the drawer without causing additional collisions with the drawer or cabinet. The robotic arm must then reposition to the handle, push it smoothly to close the drawer, and retreat a specified distance from the drawer, ensuring the arm remains stable and intact throughout the entire process without colliding with the drawer or cabinet.

### C.1.3 Level 3: Generalizable Short Manipulation Tasks.

1. **Pick**

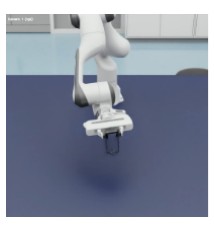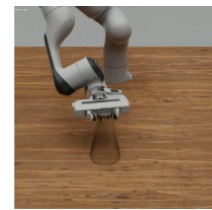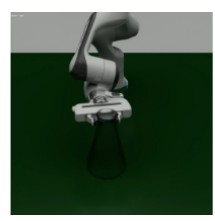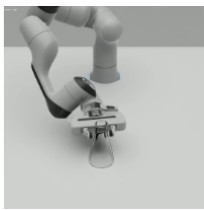

- **Description:** Consistent with the Level 1: Pick task, this task primarily focuses on generalizing across different types of chemical containers and desktop materials to evaluate the generalization capability of the policy, providing more diverse and generalized task data.

- **Success Criteria:** Aligned with Level 1:Pick task

2. **Press**

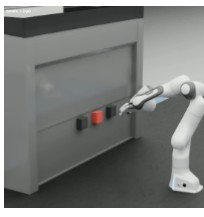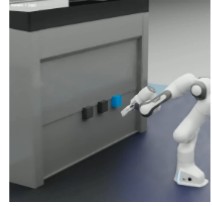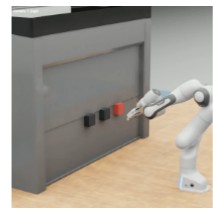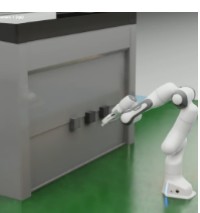

- **Description:** Consistent with the Level 1: Press task, this task primarily focuses on generalizing across different button colors, button positions, and desktop materials to evaluate the generalization capability of the policy, providing more diverse and generalized task data.

- **Success Criteria:** Aligned with Level 1:Press task

3. **Transport Beaker**

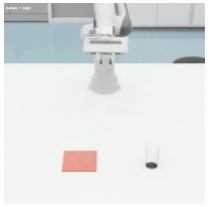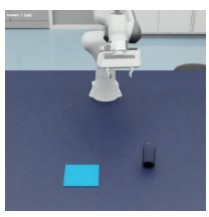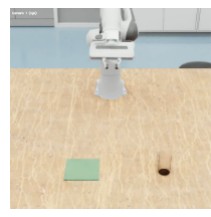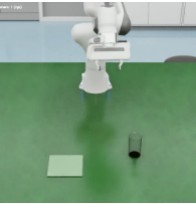

- **Description:** Consistent with the Level 2: Transport Beaker task, this task primarily focuses on generalizing across different target locations and desktop materials to evaluate the generalization capability of the policy, providing more diverse and generalized task data.
- **Success Criteria:** Aligned with Level 2:Transport Beaker task

4. **Heater Beaker**

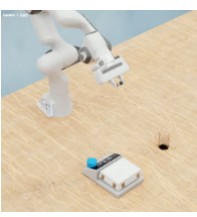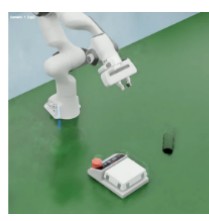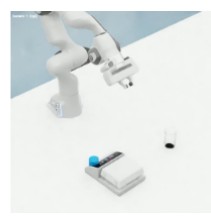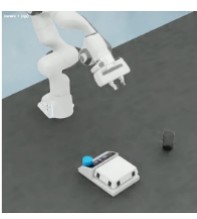

- **Description:** Consistent with the Level 2: Heater Beaker task, this task primarily focuses on generalizing across different target button colors and desktop materials to evaluate the generalization capability of the policy, providing more diverse and generalized task data.
- **Success Criteria:** Aligned with Level 2:Heater Beaker task

### C.1.4 Level 4: Long-Horizon Manipulation Tasks.

1. **Clean Beaker**

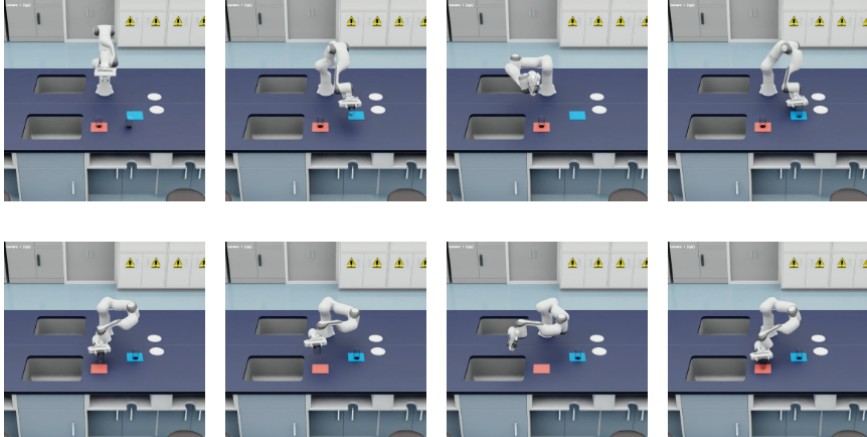

- **Description:** The Clean Beaker task represents a high-complexity robotic manipulation sequence composed of seven atomic actions—Pick, Pour, Place, Pick, Shake, Pour, and Place—executed in a precise temporal order to simulate a beaker cleaning procedure. The robot begins by performing a Pick action to grasp Beaker 2, followed by a Pour to transfer its contents into Beaker 1, and a Place to return Beaker 2 to its original position. The robot then executes another Pick action to grasp Beaker 1, performs a Shake to simulate mixing or cleaning, continues with a second Pour into a designated target beaker, and concludes with a final Place to return Beaker 1. Throughout the process, the robot must maintain accurate positioning, stable handling, and smooth transitions between actions to ensure fluid transfer integrity and task robustness.
- **Success Criteria:** The Clean Beaker task is considered successful if all seven atomic actions—Pick, Pour, Place, Pick, Shake, Pour, and Place—are executed in correct

sequence with precise control and without causing instability or spillage. Specifically, Beaker 2 must be securely grasped and lifted without tipping, its contents must be accurately poured into Beaker 1 without overflow or dripping, and it must be returned to its original position upright and stable. Subsequently, Beaker 1 must also be picked up without disturbance, shaken with controlled motion simulating realistic mixing, and its contents must be successfully poured into the designated target beaker with no leakage. Finally, Beaker 1 must be placed back at its initial location in an upright and stable orientation. Throughout the process, all containers must remain intact and upright when released, and no unintended collisions, spills, or drops should occur.

2. **Drying Beaker**

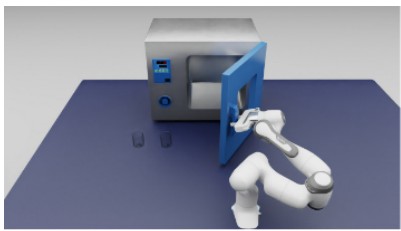
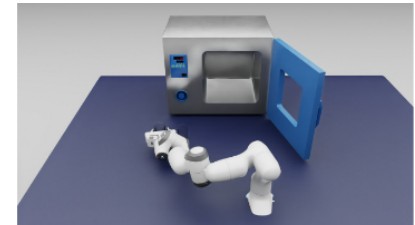
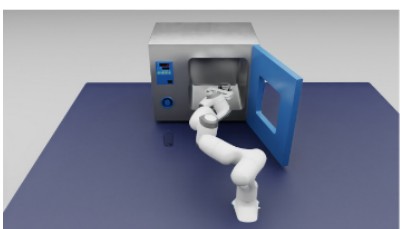
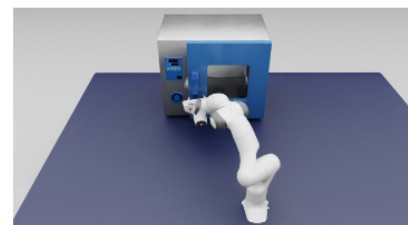

- **Description:** The Drying Beaker task represents a high-complexity robotic manipulation sequence composed of six atomic actions—Open, Pick, Place, Pick, Place, and Close—executed in a strict temporal order to simulate the process of transferring beakers into a chemical drying cabinet. The task begins with the robot performing an Open action to unlock and open the cabinet door, followed by a Pick action to grasp Beaker 2 and a Place action to accurately position it inside the cabinet. The robot then repeats the Pick and Place actions for Beaker 1, ensuring both beakers are correctly arranged for drying. Finally, the robot performs a Close action to securely shut the cabinet door. Throughout the task, the robot must ensure precise manipulation, maintain stability of the beakers during transfer, and handle the cabinet interface accurately to complete the operation reliably and safely.

- **Success Criteria:** The Drying Beaker task is considered successful if all six atomic actions—Open, Pick, Place, Pick, Place, and Close—are executed in the correct sequence with accurate control and no unintended disturbances. Specifically, the cabinet door must be fully and stably opened without obstruction during the Open action. Both Beaker 2 and Beaker 1 must be securely grasped without tipping or slipping, transported safely, and placed upright within the designated positions inside the chemical cabinet without collision or misalignment. The final Close action must result in the cabinet door being properly shut and latched. Throughout the process, all objects must remain undamaged and stable, with no drops, tilting, or contact between the beakers or with the cabinet structure that could compromise the safety or correctness of the operation.

### C.1.5 Level 5: Mobile Manipulation Tasks.

1. **Navigation and Manipulation**

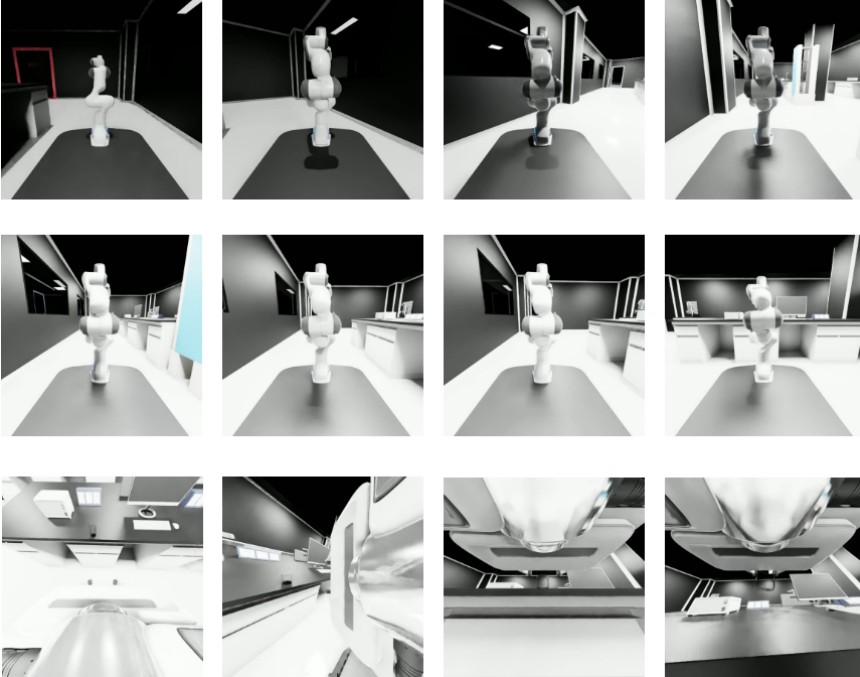

- **Description:** The Navigation and Manipulation task represents a compound robotic operation that integrates autonomous mobility with precise object interaction, combining high-level navigation with low-level manipulation. The task begins with the mobile robot and the target beaker being randomly initialized at available positions within the laboratory workspace. The robot must autonomously plan and execute a navigation trajectory to reach a position near the target beaker, ensuring obstacle avoidance and smooth traversal. Upon approaching the beaker, the robot performs a controlled orientation adjustment to face the object and halts navigation. Subsequently, the onboard manipulator executes a Pick action to securely grasp the beaker. Throughout the navigation phase, the robot must maintain full collision avoidance with static and dynamic obstacles, while the manipulation phase requires stable and precise object handling. Successful completion demands seamless integration of perception, motion planning, and control across both navigation and manipulation modules.

- **Success Criteria:** The Navigation and Manipulation task is considered successful if the robot completes all phases—navigation, orientation adjustment, and manipulation—with precise control and without unintended incidents. Specifically, during navigation, the robot must avoid any collisions with surrounding obstacles or environmental structures, following a smooth and feasible planned path. The robot must arrive steadily at a position near the target beaker and execute a controlled turn to face the beaker. Subsequently, the manipulator must perform a secure grasp of the beaker without causing it to slip or fall. Throughout the entire process, no collisions, drops, or instability of the beaker are permitted, ensuring safe and reliable task execution.

## C.2 Navigation Details

**Sampling Valid Navigation Paths.** In navigation tasks, the starting position of the agent and the target object's location are sampled, ensuring solvability—i.e., a collision-free path must exist from the starting position to the target location. First, we generate an occupancy map for each scene. We define appropriate collision volumes for the objects in the scene and then project all objects with heights between [0.1, 1.6] meters onto the ground plane, marking these areas as occupied grids. Grids outside the floor are designated as undefined regions, both of which are impassable. The remaining unoccupied grids represent passable areas. Each pixel in the occupancy map corresponds to a unit length of 0.5 meters.

Next, using the occupancy map, we generate collision-free paths from randomly sampled locations to object positions. The collision detection radius is set to 60 cm. To ensure task validity, these paths are used as the ground-truth paths for navigation tasks.

**Robot.** We employ a mobile manipulation robot composed of the Clearpath Robotics Ridgeback base integrated with a Franka Emika Panda arm for our navigation and manipulation tasks. In the Isaac Sim, we simplify the Ridgeback base to a planar holonomic robot capable of horizontal movement on the ground, allowing for efficient simulation of navigation behaviors while preserving the core manipulation capabilities of the system.

## D  Impact Statement

This paper introduces LabUtopia, a simulation and benchmarking suite aimed at advancing the development of embodied agents for scientific laboratory tasks. Its societal impact lies in its potential to fundamentally transform laboratory automation, enabling faster and more accessible scientific discovery in fields such as chemistry and materials science. By providing a high-fidelity platform for the training and evaluation of agents, LabUtopia can reduce the reliance on resource-intensive physical experiments, allowing institutions with limited infrastructure to participate in advanced research and thus enhancing research inclusivity. Ethically, LabUtopia simulates robotic operations in laboratory experiments, reducing the experimental risks faced by human researchers. This work aligns with the goals of promoting scientific innovation, improving safety, and fostering equitable access to cutting-edge research tools.

