# OpenReview forum: "LabUtopia: High-Fidelity Simulation and Hierarchical Benchmark for Scientific Embodied Agents"
_NeurIPS.cc/2025/Datasets_and_Benchmarks_Track — NeurIPS 2025 Datasets and Benchmarks Track poster_

### Official Review · Reviewer_5G6a · 2025-07-03

**Rating:** 5
**Confidence:** 4

**Summary:**

This paper introduces LabUtopia, a simulated chemistry lab environment developed using IsaacSim. The environment includes a rich set of domain-specific assets, including chemistry containers, instruments, and other lab scene elements. It supports procedural scene generation to facilitate scalable data creation. The authors develop pipelines for generating demonstration trajectories, which are then used to train and evaluate two state-of-the-art imitation learning methods: ACT and Diffusion Policy. Experimental results show that ACT performs well on short-horizon tasks, but both methods struggle on longer-horizon tasks. The inclusion of realistic chemical assets and reaction modeling contributes valuable infrastructure to studying robotic learning in scientific and laboratory contexts. The benchmark also yields insights into the limitations of current imitation learning approaches in such complex environments.

**Dataset Code Accessibility:**

Partly

**Dataset Code Comments:**

5 LabScenes on dataverse, but the huggingface link seems to be empty at the time of this review.

**Ethical Considerations:**

No, there are no or only very minor ethics concerns

**Final Justification:**

All my concerns in the rebuttal have been addressed with either clarifications or additional results. Given that not many benchmarks exist in chemistry laboratory settings, I'm convinced that this paper will be a great addition to the field.

**Limitations Weaknesses:**

- The paper lacks citation and comparison to relevant prior work, most notably:
    - Li et al., "Chemistry3D: Robotic Interaction Benchmark for Chemistry Experiments," 2024.

    Chemistry3D includes lab scenes, containers, and chemical reaction modeling. The absence of this comparison makes it difficult to assess the novelty and significance of LabUtopia’s contributions. While LabUtopia appears to improve over generic simulators, it’s unclear how it compares to more domain-specific environments.

- The procedural environment generation pipeline does not appear to reflect real-world laboratory spatial rules or usage patterns. In practice, certain instruments and containers are frequently co-located due to workflow conventions (e.g., commonly used tools placed nearby or always stored on specific shelves). Ignoring these spatial correlations may limit the simulator’s utility for studying efficient robot behavior or object navigation in realistic lab settings.
- The paper incorporates LLM-based reasoning for chemical reactions, but there is no evaluation of the model’s accuracy. While the LLM might correctly identify end products, it’s unclear whether it can handle other important aspects of chemical reactions, such as predicting heat emission or phase change.
- The framework currently lacks a task planning module that maps from a high-level experimental protocol to robot-executable plans. This capability is essential for enabling end-to-end robot lab assistants and would be particularly valuable for downstream applications.
- Although the paper claims support for deformable and fluid simulation, there is no visual evidence of this in the paper, supplementary materials, or the project website. Demonstrations of pouring and stirring involve empty containers, and task definitions (e.g., for stirring) focus solely on spatial positioning and orientation, without requiring or verifying successful manipulation of liquids.
- Physical realism remains unclear. It appears that some interactions may use non-physical "magic hand" mechanisms. For instance, in the stirring task video, the gripper visibly jerks and rotates after grasping the glass rod, yet the rod remains rigidly fixed in orientation, suggesting the absence of rigid-body contact dynamics. The paper should clarify whether grasping is implemented through physics-based contact or kinematic attachment.
- The evaluation of long-horizon tasks is incomplete. The paper omits reporting overall task success rates for Level 4, and completely excludes Level 5. It is unclear whether success rates are near zero, or whether the experiments were not conducted. Reporting these would improve the completeness and transparency of the benchmark suite.

**Strengths Contributions:**

- The paper is very well written and clearly presented. The visualizations are polished and effectively communicate key ideas and results. The overall presentation quality is commendable.
- LabUtopia provides a complete and modular pipeline, covering procedural scene generation, demonstration collection, and robot policy evaluation, along with baseline performance results.
- The simulation includes an extensive and realistic set of chemistry lab assets, such as containers and instruments. These assets are highly relevant for enabling research in robotic laboratory assistance and are modeled with high fidelity.
- The LabBench task suite offers a thoughtful decomposition of lab activities, ranging from short-horizon skill primitives to long-horizon, composed tasks. This enables more granular evaluation of robotic policies across varying levels of complexity.
- The paper presents a thorough empirical evaluation of two state-of-the-art imitation learning methods (ACT and Diffusion Policy) across diverse scenarios, including generalization to novel objects, offering valuable insights into current limitations and challenges in long-horizon manipulation.

---

> ### Author Rebuttal · Authors · 2025-07-31
>
> Dear Reviewer 5G6a,
>
> We sincerely thank you for your constructive and encouraging feedback. We truly appreciate your recognition of LabUtopia’s contributions, including its well-structured and comprehensive pipeline, extensive and realistic asset collection, and the utility of our benchmark suite.
>
> We have already incorporated several improvements based on your suggestions and will include these revisions in the updated manuscript.
>
> > **Q1: Lack of citation and comparison to relevant prior work.**
>
> A1: Thank you for the helpful suggestion. We will include both the citation and a detailed comparison to Chemistry3D in the revised version. Compared to Chemistry3D, LabUtopia offers a more comprehensive and versatile platform:
> 1. It provides a significantly larger and more diverse set of assets (more than 10 times), enabling a wider range of manipulation tasks.
> 2. LabUtopia supports mobile robots capable of performing mobile-manipulation tasks, whereas Chemistry3D is limited to desktop-based manipulation.
> 3. It includes a fully developed API for data collection and policy training, which is not available in Chemistry3D.
>
> > **Q2: Procedural environment generation lacks real-world spatial rules.**
>
> A2: We appreciate the reviewer’s valuable insight. Our pipeline enables flexible and scalable scene generation, and we are extending it with a rule-based placement module that encodes co-occurrence priors and functional zones to enhance spatial realism. Specifically, we define spatial templates (e.g., water-related tools near sink, chemical storages near fume hood) based on domain heuristics extracted from lab guidelines. These constraints are enforced during sampling via a soft energy-based layout optimization, ensuring both diversity and functional plausibility. This extension is currently under development and will be included in the revision.
>
> > **Q3: No evaluation of LLM’s accuracy for chemical reactions.**
>
> A3: Thank you for this insightful question. We conducted a two-tiered evaluation to ensure the chemical validity (shown in Tab. 1):
>
> 1) Level 1: Basic reactions: We evaluated well-established reactions (including phase change prediction) collected from PubChem [1], and found that GPT-4o-mini consistently generated chemically accurate outcomes aligned with ground-truth results.
>
> 2) Level 2: Mechanistic reasoning and quantitative prediction: We tested more complex scenarios from Chembench [2], including tasks involving heat emission. Our modular pipeline enables flexible replacement to employ stronger LLM (like GPT-4o) to further enhance reasoning for thermodynamic and physical property predictions.
>
> Table 1: Validation of chemical correctness Across Models.
>
> |  Evaluation Level  | Task Description | GPT-4o-mini | GPT-4o | Deepseek R1 |
> |--------------------|------------------|-------------|--------|-------------|
> | Level 1: Basic Reactions| Common, well-established reactions     | 95.5 | 99.5 | 99.5 |
> | Level 2: Chembench Tasks| Mechanistic reasoning and calculations | 70.1 | 99.3 | 90.7 |
>
> > **Q4: Lacks a planning module that maps from experimental protocol to robot-executable plans.**
>
> A4: Thank you for the comment. We would like to clarify that our benchmark already includes a complete pipeline from high-level experimental protocols to robot-executable actions. Specifically, Level 1 defines a set of atomic actions (e.g., pick, place, stir) that the robot can execute. Levels 2–3 compose short action sequences by chaining these primitives to perform simple tasks. For Levels 4–5, we provide an LLM-API interface (with designed prompt) that translates experimental protocols into sequences of atomic actions by leveraging a learned action library. This hierarchical design enables effective end-to-end task execution and facilitates research on planning and execution in complex lab settings.
>
> Prompt Sketch (due to limited space):
> ```
> You are a robot action planning assistant for an acid-base neutralization experiment in a simulated environment. Available skills: walk <obj>, pick <obj>, pour <obj> <obj> <req>, open <obj>, close <obj>, stir <obj>, place <obj> <obj>, monitor <obj>, press <obj>.
>
> **Step 1: Task Decomposition**
> User request: '{request}'
> Break down the task into subtasks, focusing on chemical procedures. Specify quantities for chemicals/materials and phenomena to observe. Ignore safety precautions.
>
> Output format:
> ===BEGIN===
> 1) Subtask 1
> 2) Subtask 2
> ...
> ===END===
>
> **Step 2: Subtask Processing**
> Subtask: '{subtask}'
> Environment: '{environment_obs}'
> Refine the subtask into concise actions for a two-arm robot. Use object IDs (e.g., hydrochloric_acid(id:338)) and requirements (e.g., quantities). Actions must be from the skill list and respect the two-arm limit.
>
> Output format:
> ===BEGIN_SUBTASKS===
> 1) [First action]
> 2) [Second action]
> ...
> ===END_SUBTASKS===
>
> **Step 3: Skill Mapping**
> Subtask actions: '{subtask_detail}'
> Map each action to a skill from the list, ensuring feasibility for a two-arm robot. Include parameters (e.g., quantities, durations) and use object IDs.
>
> Output format:
> ===BEGIN_SKILLS===
> 1) [Skill for action 1]
> 2) [Skill for action 2]
> ...
> ===END_SKILLS===
> ```
>
> > **Q5: Lack of deformable/fluid simulation in demonstrations.**
>
> A5: Thank you for this suggestion. To ensure clarity in task demonstration, we initially omitted complex and time-consuming simulations in the videos. The website has now been updated to include new demonstrations covering fluid, chemical-reaction simulations.
>
> > **Q6: Non-physical "Magic Hand" mechanisms.**
>
> A6: Thank you for pointing this out. We emphasize that most of our interactions are implemented by physics-based contact dynamics to ensure physical realism. The only exception is the stirring task, where we employed a kinematic attachment (“magic hand”) mechanism to simplify the grasping of the glass rod, as the primary focus was on evaluating the agent's spatial positioning rather than contact-rich manipulation. We will explicitly clarify the design choice in the revised version.
>
> > **Q7: Incomplete evaluation of Level 4 & 5 tasks.**
>
> A7: We apologize for the confusion. In Tab. 3 of the paper, we report the success rates for the single policy (SP) and combined policy (denoted as “A”), with A7 reflecting the final success rate. We will make this description clearer in the revision.
>
> As for Level 5, we did conduct experiments and our results indicate that these tasks remain extremely challenging under current policy capabilities. Success requires high-precision coordination between the mobile base and the robotic arm to accurately reach and manipulate targets. All learning-based policies we evaluated achieved a success rate of **0%** on Level 5.
>
> We will explicitly report these results in the revision to improve the completeness of the benchmark suite.
>
> > **Q8: Empty HuggingFace link.**
>
> A8: Following the conference requirements, we have provided a complete preview dataset access in the Harvard Dataverse. We will upload the data to HuggingFace and make it publicly available in the subsequent open-source release.
>
> [1] Kim, Sunghwan, et al. "PubChem substance and compound databases." Nucleic acids research 44.D1 (2016): D1202-D1213.
>
> [2] Zhang, Di, et al. "Chemllm: A chemical large language model." arXiv preprint arXiv:2402.06852 (2024).

---

> > ### Comment · Reviewer_5G6a · 2025-08-05
> >
> > I thank the authors for the clarifications and new results. My concerns have been adequately resolved. I especially appreciate the LLM evaluation and the additional videos added for fluid simulation. The example for grounding experiment protocol to robot actions is also helpful. I have raised my rating accordingly.
> >
> > Even with the large number of reviewers, the authors did a great job responding to each and every comment and suggestion. You have my personal salute [hats off].

---

> > > ### Author Response · Authors · 2025-08-06
> > >
> > > Thank you for your recognition and positive feedback. Your insightful comments will significantly strengthen our paper, and we sincerely appreciate your favorable evaluation and the increase in your rating. Regardless of the number of reviewers, we value every suggestion and do our utmost to address each one carefully in order to further improve the quality of our work.
> > >
> > > Best regards,
> > >
> > > The Authors

---

### Official Review · Reviewer_ibFk · 2025-07-04

**Rating:** 4
**Confidence:** 4

**Summary:**

LabUtopia is simulation benchmarking platform tailored to laboratory environments, enabling embodied agents to perceive complex physico-chemical transformations and tackle long-horizon tasks. It combines LabSim (multi-physics simulator), LabScene (procedural scene generator), and LabBench (five-level hierarchical benchmark) with 30 tasks and 200+ instrument and scene assets. Together, these components provide a testbed for evaluating agents that integrate perception, planning, and control in scientific workflows.

**Dataset Code Accessibility:**

Yes

**Ethical Considerations:**

No, there are no or only very minor ethics concerns

**Final Justification:**

The authors address some of my concerns, but the key issue, that the scientific simulation is not realistic, cannot be resolved.

**Limitations Weaknesses:**

1. Most available tasks focus on low-level motor skills rather than science-centric reasoning.
2. The subset of tasks that reflect laboratory research is too small.
3. The simulator cannot yet model fine-grained scientific phenomena, for example, detailed chemical reactions, which may need the operation to be very accurate.
4. The environment omits accurate reward signals, which are crucial for both training agents and evaluating their performance.

**Strengths Contributions:**

1. The authors introduce LabUtopia with 30 benchmark tasks and 200 + instruments, containers, and lab scenes foster broad generalization instead of overfitting to a handful of setups.
2. The benchmark provides five difficulty tiers, from low-level atomic actions to full mobile-manipulation protocols, let researchers measure progress incrementally and target specific skill gaps.

---

> ### Author Rebuttal · Authors · 2025-07-31
>
> Dear Reviewer ibFk,
>
> Thank you for your effort and useful advices in review process. We appreciate your recognition of the contributions made by our work, particularly the comprehensive task and scene diversity in LabUtopia (30 benchmark tasks and over 200 instruments), and the hierarchical benchmark design that spans five levels of difficulty.
>
> We have already incorporated several improvements based on your suggestions and will include these revisions in the updated manuscript.
>
> > **Q1: Insufficient focus on science-centric reasoning.**
>
> A1: Our benchmark is designed to span the full spectrum from low-level atomic manipulations to high-level scientific reasoning. Levels 1-3 concentrate on the fundamental atomic operations that underpin day-to-day laboratory practice, whereas Levels 4 and 5 capture increasingly sophisticated reasoning tasks. We are also extending the benchmark with richer, more complex experimental scenarios to further strengthen its coverage of science-centric reasoning.
>
> > **Q2: Limited laboratory research tasks.**
>
> A2: Thank you for your advice. The tasks included in our benchmark, such as pouring, stirring, and transporting beakers, are fundamental actions integral to laboratory research. As these actions occur frequently in real experiments, they represent core manipulation and interaction skills essential for conducting scientific experiments. In addition, we further incorporate 1) Iodine Clock Reaction, designed to evaluate the model’s ability in chemical reaction reasoning. 2) Laboratory Bench Cleaning task aimed at testing the free action squence manipulation:
>
> Task 1 description:
> Laboratory bench cleaning task consists of three subtasks: close the drying oven, pour liquid, and moving the beaker. These three tasks can be performed independently. During data collection, the execution order is randomly chosen. We used DP and ACT for training on 100 demonstrations for 100 epochs.
>
> Evaluation Results:
>
> Table 1: Performance comparison of different models on free action sequence task.
>
> | Model | Close drying oven | Pour liquid | Move the beaker | Overall success rate |
> |-------|-------------------|-------------|-----------------|----------------------|
> | ACT   |        48%        |     36%     |       40%       |        6%            |
> | DP    |        24%        |     32%     |       16%       |        1%            |
>
> Task 2 description:
> Three small beakers are randomly placed on the table—two containing pure water and one containing a 1% starch solution. Another beaker holds a mixed solution of 0.05 mol/L potassium iodide (KI) and 0.01 mol/L iodine (I₂). The policy is tasked with sequentially pouring the contents of each small beaker into the beaker with the mixed solution, without prior knowledge of which beaker contains water or starch. Each time a solution is added, an LLM-based chemistry reaction engine is invoked to update the mixture's properties. When a color change reaction is detected, the robotic arm presses a button to indicate the end of the experiment.
>
> We evaluate the experiment using two success rates:
>
> 1. Action success rate: This measures whether all required actions are performed correctly—specifically, at least one pouring action followed by pressing the button to end the experiment—regardless of whether a color change occurs in the solution.
> 2. Strict success rate: This requires that, after a color change is observed, no additional solutions are poured into the beaker, and the button is successfully pressed to indicate the end of the experiment.
>
> Evaluation Results:
>
> Table 2: Performance comparison of different models on Level-4 Iodine Clock Reaction.
> | Model |  Action success rate | Strict success rate  |
> |-------|----------------------|----------------------|
> | ACT   |      13%             |         4%           |
> | DP    |       6%             |         2%           |
>
> We plan to further expand the task set in future iterations to include more diverse and complex experimental scenarios, enhancing the simulator’s capabilities in scientific experiment simulation.
>
> > **Q3: Modeling fine-grained scientific phenomena.**
>
> A3: Thank you for your insightful comment. Precisely simulating detailed chemical reactions remains a fundamental challenge for both the chemistry and embodied intelligence communities, which often involve computationally intensive methods such as quantum chemistry or molecular dynamics. Such methods, in our opinion, are beyond the scope of this work. LabUtopia makes a preliminary attempt at chemical simulation by providing a framework to test the perception and reasoning capabilities of laboratory embodied agents. While our current focus is on broad physical interactions, we recognize the need for precise chemical modeling. This is a critical future direction for our work and requires collaborative efforts from researchers across multiple domains.
>
> > **Q4: Lack of accurate reward signals.**
>
> A4: Since our benchmark primarily tests imitation learning methods, we did not define detailed reward functions for each task and instead adopted a default binary "success-or-not" metric for the tasks. However, LabSim's underlying framework exposes sufficient information for users to construct dense rewards, including physical states (6D pose, velocity, force/torque, robotic arm state, etc.) and chemical states (reactions, color changes, precipitate mass, etc.) returned by the chemical engine. This allows users to construct custom dense or continuous reward functions as needed for reinforcement learning or more comprehensive evaluation.

---

### Official Review · Reviewer_385p · 2025-07-05

**Rating:** 5
**Confidence:** 3

**Summary:**

The paper introduces LabUtopia, a simulation and benchmarking suite for evaluating embodied agents in laboratory settings. It contains:
- `LabSim` simulates the multi-physics and chemically meaningful interactions. Several physical simulation methods are used to simulate the physical interactions among assets. The chemical processes are generated by integrating a curated knowledge base with a reasoning model (a large language model).
- `LabScene`  is a scalable dataset of laboratory object and scene assets. Candidate scenes are collected from design websites. A a high-quality subset is selected with input from domain experts in chemistry and physics, and with additional post-processing steps to make them suitable for the dataset.
- `LabBench` is a hierarchical benchmark that spans five levels of increasing task complexity, from atomic actions to long-horizon mobile manipulation tasks.

The benchmark is used to evaluate two representative models: ACT and Diffusion Policy. Two "camera" are used to convert the simulation environment into image-based inputs for the models. Evaluation is performed in real-time within the simulated environment. The two models are evaluated across different levels of the tasks. The paper also provides performance comparison and discussion of the performance difference.

**Additional Feedback:**

> The chemical process is generated by leveraging a large language model (GPT-4o mini) to reason about potential chemical processes and infer corresponding transformations, including color changes, product formation.

* How was the correctness of these generated chemical processes evaluated to ensure they result in a realistic and reliable dataset?

> iii) In model selection, we failed to test vision-language-action (VLA) models.

* Could you add a bit more details about why this failed?

**Dataset Code Accessibility:**

Yes

**Dataset Code Comments:**

The dataset and code are provided, with detailed steps to run the code.

**Ethical Considerations:**

No, there are no or only very minor ethics concerns

**Final Justification:**

The authors' response has addressed the questions that I initially had, so changing the score from 4 to 5.

**Limitations Weaknesses:**

I just have some minor questions in the "Additional Feedback" section below.

**Strengths Contributions:**

* The proposed dataset contains a fair amount of complexities and diversities in terms of assets, scenes, and task types. The steps of the dataset collection and generation process are described in details.

* The paper provides a thorough evaluation for the two baseline models on different levels of tasks. Besides, the paper includes insightful analysis and discussion of the performance differences across task levels and models.

---

> ### Author Rebuttal · Authors · 2025-07-31
>
> Dear Reviewer 385p,
>
> We sincerely thank the reviewer for the constructive and thoughtful feedback. We greatly appreciate your recognition of LabUtopia’s comprehensive simulation framework, the diversity and quality of the asset and scene dataset, and the detailed evaluation across multiple task levels. Your positive assessment of our benchmark design and analysis is highly encouraging.
>
> We fully agree with your suggestions and have already incorporated several improvements based on your comments:
>
> > **Q1: Chemical process correctness evaluation.**
>
> A1: Thank you for this insightful question. We conducted a two-tiered evaluation to ensure the chemical validity, as summarized in Tab. 1:
>
> 1) Level 1: Basic reactions with well-established outcomes from PubChem [1]. The reaction reasoning module with GPT-4o-mini can generate outputs that are chemically accurate and aligned with ground-truth outcomes. This covers fundamental organic and inorganic reaction types frequently found in textbooks and databases.
>
> 2) Level 2: Mechanistic reasoning and calculation problems from Chembench [2]. For more frontier tasks, Our modular pipeline enables flexible replacement to employ stronger LLM (like GPT-4o) to further enhance chemical process correctness.
>
> Table 1: Validation of chemical correctness Across Models.
>
> |  Evaluation Level  | Task Description | GPT-4o-mini | GPT-4o | Deepseek R1 |
> |--------------------|------------------|-------------|--------|-------------|
> | Level 1: Basic Reactions| Common, well-established reactions     | 95.5 | 99.5 | 99.5 |
> | Level 2: Chembench Tasks| Mechanistic reasoning and calculations | 70.1 | 99.3 | 90.7 |
>
> > **Q2: The reason for the omission of VLA model testing.**
>
> A2: Thank you for highlighting this point. The omission was primarily due to the initial focus on pure imitation-learning policies, which aligned with the benchmark’s design to test baseline performance on laboratory tasks. To achieve more comprehensive evaluations, we have supplemented our experiments with the widely-adopted VLA model, Pi0 3], evaluated on Level-3 tasks.
>
> The results in Tab. 2 reveal two notable observations:
> 1. Pretrained VLA models, after fine-tuning, do not always outperform models trained from scratch in our tasks.
> 2. VLA models show negligible performance degradation when evaluated on out-of-distribution visual inputs or novel materials.
>
> These findings underscore both the complexity of LabUtopia tasks and the potential of VLA models for robust embodied AI. We plan to expand our evaluations further in future work.
>
> Table 2: Performance comparison on Level-3 Tasks (ID/OOD success rate %).
> | Task              | Pi0  | ACT  | DP   |
> |-------------------|------|------|------|
> | Pick              | 83.3/85.8 | 81.7/71.7 | 53.3/41.7 |
> | Press             | 92.5/89.1 | 98.3/96.7 | 81.7/31.6 |
> | Open Door         | 51.6/53.3 | 73.3/65.0 | 63.3/58.3 |
> | Pour Liquid       | 40.0/38.3 | 75.0/65.0 | 46.6/31.6 |
> | Heater Beaker     | 89.1/86.7 | 86.7/80.0 | 21.6/8.3  |
> | Transport Beaker  | 86.7/88.3 | 77.5/73.3 | 67.5/15.0 |
>
> [1] Kim, Sunghwan, et al. "PubChem substance and compound databases." Nucleic acids research 44.D1 (2016): D1202-D1213.
>
> [2] Zhang, Di, et al. "Chemllm: A chemical large language model." arXiv preprint arXiv:2402.06852 (2024).
>
> [3] Black, Kevin, et al. "$\pi_0$: A Vision-Language-Action Flow Model for General Robot Control." arXiv preprint arXiv:2410.24164 (2024).

---

### Official Review · Reviewer_mfSr · 2025-07-08

**Rating:** 5
**Confidence:** 4

**Summary:**

LabUtopia is a simulation environment and benchmark for embodied agents placed in a laboratory setting. It includes a physics simulation part, a procedural tool for generating laboratory scenes, and a benchmark of over 30 tasks of varying difficulty. The authors include an evaluation of the benchmark on two baseline models.

**Additional Feedback:**

- How many scene assets are included in the environment? The paper mentions 1000 candidate scenes, how many were subselected?
- As it is not presented in the benchmark, I am wondering how are the chemical reactions simulated. In the real world, they happen over time. Is it the same in simulation, or is it instantaneous state change? Notably, it would be another example of an interesting task in which reasoning depends on assessing the reaction completion based on reaching the desired state.
- I wonder if the main difficulty of the performance of the models comes from the transparency of the task target objects captured in low resolution. It may be worth exploring substituting the materials for those to investigate whether the visual part of the recognition poses the biggest challenge.

**Dataset Code Accessibility:**

Yes

**Dataset Code Comments:**

Both the dataset and code for running the env are released and available.

**Ethical Considerations:**

No, there are no or only very minor ethics concerns

**Final Justification:**

I am satisfied with the rebuttal. I believe all my concerns were addressed (including the main one of showing the unique strength of the benchmark - chemical reactions - within the evaluation tasks). Similarly, the responses to other reviewers were convincing and extensive.

**Limitations Weaknesses:**

There are several limitations of the work, mostly stemming from a lack of description with respect to details of the system, and a lack of benchmarking tasks showcasing the claimed capabilities of the system.
- The main limitation I notice regarding the proposed environment is a mismatch between the claims made about the simulation and the offered benchmark. The authors put some emphasis on the inclusion of the chemical reactions in the environment (i.e. changing the state, e.g. colour when undergoing reaction). However, looking at the list of tasks in the supplementary material, it seems that none of that is actually included in the benchmark (similarly, not in any example on the website). If this is not included, isn't the benchmark rather similar to existing Task and Motion Planning benchmarks (e.g. Ai2Thor, ManipulaThor, etc.) but set in a laboratory environment. The inclusion of chemical reactions in the simulation opens a lot of interesting possibilities for benchmarking that are left unexplored, e.g. pour fluid from beaker to beaker until the colour changes.
- In a similar manner, I am not fully convinced about the success criteria of some of the tasks. In e.g. pouring (see supplementary material B.1.2 1. Pour liquid, it does not seem that there is a liquid transfer between the containers. Following that, my suspicion is that the success criterion is based on the geometrical movement of the source container. However, there should be a measure of success based on the amount of liquid transferred to the destination container, as this seems like a constraint important in a laboratory setting (lack of spillage), especially given that in a simulation with various fluids included, they may be of varying viscosity, thus having to be poured differently. Similarly, in tasks with several objects, the state of the objects not included in the task itself should affect the evaluation of the task, namely, if other objects are knocked over, it is important to reflect this in the score. The same considerations can be applied to any of the proposed tasks that can include fluids. In shaking action, one could pay attention to the density of the fluid and how full the beaker is, as it could affect the 'amplitude' of the shaking.
- As it is a robotic benchmark, I would like to see a more structured description of the inputs and available observations. It seems that the input is the cartesian coordinates of a desired pose of the end effector. However, are the images the only feedback? It would be important to include force feedback in real life, especially when manipulating glass objects.
- It would be good to include more information on the camera placement. It is worth noting that many common manipulator setups include a camera attached to the robot's wrist.
- Regarding complex tasks in the benchmark, it seems that non-atomic tasks are always composed of the same sequence of atomic actions. This makes it very easy for the reasoning system to overfit on the task planning level and weakens the reasoning part of benchmarking.
- There is no evaluation on Level 5 tasks, making it hard to judge the reasonability of that part of the benchmark.
- It seems that the major benefits of the simulator come from Isaac Sim itself (fluids, deformable objects), it would be worth clarifying which parts of the simulation environment are contributions.
- It would be good to include more than 1 robot and gripper.

In summary, the simulation environment sounds like a good idea, but the corresponding benchmark lacks a bit of maturity, and the main advantages of the environment are not showcased and thus proven to be effective.

**Strengths Contributions:**

- The paper introduces a navigation and planning benchmark set up in a laboratory environment. It is an interesting idea, well motivated by the inclusion of automated machines in such environments.
- The inclusion of fluids and deformable objects in simulation is a good idea, well suited to the lab environment.
- Simulation of chemical processes in a robotics simulator is a novel and interesting idea, well-motivated by real-world possibilities.
- Scene and object selection and curation look impressive, and the examples presented in the paper, supplementary and webpage look appropriate for the laboratory environment.
- The approach to environment generation sounds sensible, and the presented examples show viable settings.
- I appreciate the complexity-level structure of the benchmark, incorporating tasks designated as easier and harder.
- The choice of baselines seems reasonable.

---

> ### Author Rebuttal · Authors · 2025-07-31
>
> Dear Reviewer mfSr,
>
> Thank you for your thoughtful and constructive feedback. We sincerely appreciate your comments, which will help us improve the clarity and completeness of the paper.
>
> We have already incorporated several improvements based on your suggestions and will include these revisions in the updated manuscript.
>
> > **Q1: Mismatch between chemical simulation claims and benchmark tasks.**
>
> A1: Thank you for this comment. We have added a level-4 task **Iodine Clock Reaction**, which explicitly involves chemical reactions and dynamic color changes. Corresponding demonstration videos with full simulation are now available on the project website.
>
> Task description: Three small beakers are randomly placed on the table—two containing pure water and one containing a 1% starch solution. Another beaker holds a mixed solution of 0.05 mol/L potassium iodide (KI) and 0.01 mol/L iodine (I₂). The policy is tasked with sequentially pouring the contents of each small beaker into the beaker with the mixed solution, without prior knowledge of which beaker contains water or starch. Each time a solution is added, an LLM-based chemistry reaction engine is invoked to update the mixture's properties. When a color change reaction is detected, the robotic arm presses a button to indicate the end of the experiment.
>
> We evaluate the experiment using two success rates:
>
> 1. Action success rate: This measures whether all required actions are performed correctly—specifically, at least one pouring action followed by pressing the button to end the experiment—regardless of whether a color change occurs in the solution.
> 2. Strict success rate: This requires that, after a color change is observed, no additional solutions are poured into the beaker, and the button is successfully pressed to indicate the end of the experiment.
>
> Evaluation Results:
>
> Table 1: Performance comparison of different models on Level-4 Iodine Clock Reaction.
> | Model |  Action success rate | Strict success rate  |
> |-------|----------------------|----------------------|
> | ACT   |      13%      |  4%    |
> | DP    |6%      |  2%    |
>
> Lastly, we acknowledge that due to the slower rendering speed of fluid simulation, our demo videos primarily highlight task categories rather than detailed chemical effects. We have prepared a full set of rendered videos and interactive environments showcasing these reaction-centric tasks, to better demonstrate LabUtopia’s distinct contributions beyond traditional household simulators like AI2-THOR or ManipulaTHOR.
>
> We hope these additions effectively address your concern and demonstrate the unique value of LabUtopia in laboratory robotic benchmarks.
>
> > **Q2: Clarify the success criteria for tasks.**
>
> A2: Thank you for your valuable suggestions. Our current success criteria is based on hard-coded geometric standards. Following your suggestion, we have revised our evaluation protocol to incorporate two universal safety and realism constraints inspired by standard laboratory guidelines:
> 1) No spillage: During pouring or transportation, the liquid must not overflow or leak.
> 2) Container tilt constraint: Containers must remain within 15° of vertical to reflect careful handling.
>
> We integrated these constraints into the evaluation of two representative tasks: "Pour Liquid" and "Transport Beaker". As shown in Tab. 2, enforcing these constraints led to a notable drop in success rates, highlighting the complexity and precision required in real lab environments.
>
> We believe these refinements can enhance the realism and rigor of our benchmark, and we will include more criteria, evaluation details, and experimental results in the revision.
>
> Table 2: Success rates before and after applying constraint.
>
> |Task| Original|With Constraint | Change  |
> |----|---------|---|---|
> |Pour Liquid|67.5%|51.6%|↓ 15.9% |
> |Transport Beaker|78.3%|70.6%| ↓ 7.7%  |
>
> > **Q3: Structured description of inputs and observations.**
>
> A3: Thank you for this suggestion. In the revised manuscript, we will add a detailed subsection specifying that inputs include the desired 6D pose of the end effector, while observations encompass RGB images from multiple cameras, and proprioceptive data. Force feedback, critical for manipulating fragile glass objects, is supported by force sensors on the robotic arm’s end effector, providing numerical data on force interactions. However, as the tested policies did not utilize force feedback as an observation input, we did not emphasize this aspect.
>
> > **Q4: Clarification of camera placement in experiments.**
>
> A4: In Section 5.1, we provide general principles for camera placement, following the original settings of policy models. To clarify further: all ACT and DP models were trained using two camera views —— one directly overhead and one diagonally facing the robotic arm. For the open task, the camera facing the arm was mounted on the arm itself due to equipment obstructing the original position. In the added VLA experiments, we consistently used the camera mounted on the robot arm's wrist. We will revise the ‘Visual Input’ section in the revision.
>
> > **Q5: Composition of complex tasks.**
>
> A5: Thank you for your suggestion. Random action sequences indeed pose a significant challenge for imitation learning, which was overlooked in our original benchmark. We have added a new task to Level-4 Laboratory Bench Cleaning task, comprising three actions, with these actions executed in a random order during data collection. Our test results are as follows:
>
> Task description:
> Laboratory bench cleaning task consists of three subtasks: close the drying oven, pour liquid, and moving the beaker. These three tasks can be performed independently. During data collection, the execution order is randomly chosen. We used DP and ACT for training on 100 demonstrations for 100 epochs.
>
> Evaluation Results:
>
> Table 3: Performance comparison of different models on free action sequence task.
> | Model | Close drying oven | Pour liquid | Move the beaker | Overall success rate |
> |-------|-------------|---------|---------|-------------|
> | ACT   |        48%        |     36%     |       40%       |        6%            |
> | DP    |        24%        |     32%     |       16%       |        1%            |
>
> Furthermore, we will provide a template to facilitate users in extending experiments more easily, enabling both community‑driven extensions and broader task diversity.
>
> > **Q6: Lack of evaluation for level-5 tasks.**
>
> A6: Thank you for the comment. Our experiments confirm that Level-5 tasks are highly challenging, due to the stringent precision requirements for the robot to accurately reach target positions and grasp objects. As a result, all tested learning-based policies achieved a **0%** success rate.
>
> To facilitate progress on these tasks, we have introduced a user-friendly path planning and chassis control tool. This tool enables rule-based navigation and supports data collection (e.g., camera input, chassis speed, robot position), helping researchers develop and evaluate more robust policies for Level 5 tasks. We will include evluation results, detailed description, and usage instructions in the revision.
>
> > **Q7: Clarify contributions of the simulator.**
>
> A7: Thank you for the helpful suggestion. We will clarify that the soft-body and fluid simulations are supported by Isaac Sim. Beyond these built-in features, our contributions include: (1) supporting a wide range of chemical substances through pre-defined physical parameters; (2) developing a chemical reaction simulation module for laboratory scenes; and (3) integrating these components into an embodied environment to support learning-based agents in tasks involving physical and chemical interactions.
>
> > **Q8: It would be good to include more than 1 robot and gripper.**
>
> A8: Thank you for your suggestion. We agree that including multiple robots and grippers wil strengthen the benchmark’s generality. We will add a UR robotic arm to the open-source version and provide a template for users to incorporate custom robotic arms.
>
> > **Q9: Clarify the number of scene assets and subselected scenes.**
>
> A9: Thank you for the suggestion. We ultimately selected 105 experimental instrument assets and 100 laboratory environment assets, all of which were made accessible to reviewers via an anonymous link during the review process. We will subsequently open-source all assets.
>
> > **Q10: Does the simulation model chemical reactions as temporal processes or as instantaneous state changes?**
>
> A10: Thank you for this insightful question. LabUtopia models chemical reactions as instantaneous state changes. Modeling the temporal dynamics of chemical reactions is extremely challenging, and currently, no simulator (in either the chemistry or embodied intelligence domains) captures the temporal progression of chemical reaction processes. We agree with the reviewer’s perspective that this is an intriguing research task in the field of embodied intelligence, which also poses significant challenges to the reasoning capabilities of models.
>
> > **Q11: Impact of low-resolution object transparency on model performance.**
>
> A11: We appreciate the reviewer’s insightful observation regarding the impact of transparent objects on model performance. Indeed, transparency poses significant challenges for the visual encoder. To test this, we replaced transparent beakers with opaque ones in the Diffusion Policy experiments on "Level-1 Pick" and "Level-2 Pour Liquid" tasks. The success rate for Pick improved from 75.0% to 89.1%, and for Pour Liquid from 50.0% to 59.1%. These results confirm that transparency adversely affects perception, which in turn impacts overall task success.
>
> Table 4:Impact of transparency on task success rates in Diffusion Policy experiments.
> | Task | Transparent Objects | Replacing with Opaque Objects | Change|
> |------|-------|-----------|--------|
> | Level-1 Pick      | 75.0% | 89.1% | +14.1%|
> | Level-2 Pour Liquid | 50.0% | 59.1% |+9.1% |

---

> > ### Comment · Reviewer_mfSr · 2025-08-03
> > **Thank you for the rebuttal**
> >
> > I thank the authors for a detailed rebuttal. I appreciate taking the suggestions into account (new task enforcing the presence of a chemical reaction, tightening the success conditions). I believe this increased the appeal of the benchmark. I intend to reflect it in my final rating.
> >
> > Finally, as small suggestions for the future, providing a tutorial/readme for the users on adding new tasks.

---

> > > ### Author Response · Authors · 2025-08-04
> > >
> > > We appreciate the reviewer’s positive feedback on our response, as well as your recognition of our enhanced benchmark. Your comments have helped us further improve our work and better articulate the significance of our contributions.
> > >
> > > We also sincerely thank you for your valuable suggestion regarding a tutorial or README for adding new tasks. We will make sure to include this in future releases.
> > >
> > > Best regards,
> > > The Authors

---

### Official Review · Reviewer_Q5Ab · 2025-07-16

**Rating:** 5
**Confidence:** 3

**Summary:**

This article introduces a high-fidelity simulation and hierarchical benchmarking platform named LabUtopia, aimed at promoting the development of embodied agents in scientific experimental environments. By integrating high-fidelity simulators, scalable scientific scenario generators, and hierarchical benchmarking, LabUtopia provides a powerful tool for training and evaluating embodied agents in laboratory settings. Despite significant progress, further research and innovation are still needed to realize a general-purpose intelligent agent suitable for scientific laboratory scenarios.

**Dataset Code Accessibility:**

Yes

**Dataset Code Comments:**

The code is available.

**Ethical Comments:**

Is it possible that the simulator could involve the simulation of some hazardous reactions?

**Ethical Considerations:**

No, there are no or only very minor ethics concerns

**Final Justification:**

Thank you very much for the authors' response. I have no further questions and will maintain my score.

**Limitations Weaknesses:**

1. Only two pure imitation-learning policies are tested; no reinforcement-learning, vision-language-action (VLA) or hierarchical planners are evaluated. Hence, difficulty claims are only partially supported.
2. Reaction reasoning is delegated to GPT-4o-mini, but prompts, caching strategy, latency and error-handling are not described; validation of chemical correctness is absent.
3. Could more dimensions of tests and analyses be included in the experiment, such as the judgment of dangerous operations?

**Strengths Contributions:**

1. Automated laboratories are an emerging, high-impact domain where long-horizon reasoning, multi-modal perception and fine manipulation meet; existing simulators are household-centric. The work fills a clear gap (Tab. 1).

2. Multi-physics + chemically-aware simulation (Sec. 3.1, Fig. 2) is novel; prior simulators ignore reaction-driven colour/state changes. Procedural scene generation with constraint-aware placement (Sec. 3.2) enables scale and diversity while remaining physically plausible. Large, curated asset library (>160 objects + 100 scenes) verified by domain experts; rare in this niche.

3. The five-level taxonomy (Fig. 3) lets researchers probe perception/control at different temporal scales and test generalisation (Level 3) separately from planning (Level 4/5). The evaluation protocol is precise (Sec. 4.2).

4. Code, assets and data are promised at a public site; Section 5.1 discloses training details, hardware, and task splits.

5. Baseline results (Tabs. 2-4) reveal that state-of-the-art imitation models that excel in kitchen/household scenes still struggle with (i) fluid-related tasks, (ii) unseen object variations, and (iii) error accumulation in long sequences—evidence that LabUtopia exposes new research challenges.

6. The paper is easy to follow; contributions are enumerated (lines 74-85) and figures effectively illustrate key ideas (Figs. 1–3).

---

> ### Author Rebuttal · Authors · 2025-07-31
>
> Dear Reviewer Q5Ab,
>
> We sincerely thank you for your constructive and encouraging feedback. We truly appreciate for your recognition of LabUtopia’s contribution on the unexplored domain of laboratory robotics, multi-physics and chemically-aware simulation, and the utility of our hierarchical five-level benchmark.
>
> We have already incorporated several improvements based on your key suggestions and will continue enhancing LabUtopia toward becoming a robust and extensible research platform for embodied AI in scientific environments.
>
> > **Q1: Extension of policy model evaluations.**
>
> A1: Thank you for this valuable suggestion. Recognizing the growing interest in Vision-Language-Action (VLA) models, we conducted supplementary experiments using a VLA-based model (Pi0[1]) on Level‑3 generalization tasks (in Tab. 1).
>
> Our results highlight that VLA models showcase stronger generalization to OOD visual inputs and novel materials. However, pretrained VLA models, after fine-tuning, do not always outperform models trained from scratch on our tasks (e.g. Pour Liquid), which further underscores the inherent complexity and challenge of these laboratory tasks.
>
> In future work, we plan to expand our evaluation suite with reinforcement learning and hierarchical planner, providing a more comprehensive evaluation under our task settings.
>
> Table 1: Performance comparison on Level-3 Tasks (ID/OOD success rate %).
> | Task| Pi0  | ACT  |  DP  |
> |-------------------|------|------|------|
> | Pick| 83.3/85.8 | 81.7/71.7 | 53.3/41.7 |
> | Press      | 92.5/89.1 | 98.3/96.7 | 81.7/31.6 |
> | Open Door  | 51.6/53.3 | 73.3/65.0 | 63.3/58.3 |
> | Pour Liquid| 40.0/38.3 | 75.0/65.0 | 46.6/31.6 |
> | Heater Beaker     | 89.1/86.7 | 86.7/80.0 | 21.6/8.3  |
> | Transport Beaker  | 86.7/88.3 | 77.5/73.3 | 67.5/15.0 |
>
> > **Q2: Supply details on reaction reasoning and validation of chemical correctness.**
>
> A2: Thank you for raising this important point. Our reaction reasoning module employs GPT-4o-mini API with designed prompts that encode the chemical context and several formatted examples to guide the simulation. The latency per API call is approximately one second. While we currently do not implement caching or error-handling, we acknowledge that these mechanisms are promising directions to enhance the module’s memory capabilities and self-correction in future work.
>
> Formatted Example:
> > Input:
> > Reactants: {"Name": "AgNO3", "State": "aqueous"}, {"Name": "NaCl", "State": "aqueous"}
> > Conditions: Room temperature, aqueous solution
> >
> > Output:
> > {
> >   "Products": [
> >     {"Name": "AgCl", "State": "solid"},
> >     {"Name": "NaNO3", "State": "aqueous"}
> >    ],
> >   "ColorChange": "White",
> >   "StateChange": "{AgCl, solid}"
> > }
>
> To validate the chemical correctness, we conducted two levels of evaluation: (1) basic reactions with well-established outcomes from PubChem[2], and (2) mechanistic reasoning and calculation problems from Chembench[3]. Our results (Tab. 2) show that GPT-4o-mini achieves near-perfect accuracy on basic reaction inferences. For more frontier research questions, our modular pipeline enables flexible replacement to employ stronger LLM (like GPT-4o) to further enhance chemical simulation fidelity.
>
> We will incorporate these detailed descriptions and validation results into the revised paper to enhance the clarity and reliability of our chemical reasoning module.
>
> Table 2: Validation of chemical correctness Across Models
>
> |  Evaluation Level  | Task Description | GPT-4o-mini | GPT-4o | Deepseek R1 |
> |--------------------|------------------|-------------|--------|-------------|
> | Level 1: Basic Reactions| Common, well-established reactions     | 95.5 | 99.5 | 99.5 |
> | Level 2: Chembench Tasks| Mechanistic reasoning and calculations | 70.1 | 99.3 | 90.7 |
>
> > **Q3: More dimensions of evaluations.**
>
> A3: Thank you for this valuable suggestion. To incorporate safety considerations, we reviewed standard laboratory guidelines and implemented two universal rules in our simulation:
> 1. Upright transport: Containers must remain within 15° of vertical during grasping and movement.
> 2. No spillage: Liquid must not overflow or leak during handling.
>
> We added these rules into evaluation on “Pour Liquid” and “Transport Beaker” tasks. Enforcing these constraints reduced the success rates of both tasks (in Tab. 3), demonstrating how safety checks expose additional challenges. We agree that safety is paramount in laboratory automation and consider the integration of safety-aware policy models to be a critical next step for embodied agents in LabUtopia.
>
> Table 3: Success rates before and after applying safety rules.
>
> | Task| Original | With Safety Rules | Change   |
> |-------------------|----------|-------------------|----------|
> | Pour Liquid| 67.5%    | 51.6%      | ↓ 15.9%  |
> | Transport Beaker  | 78.3%    | 70.6%      | ↓ 7.7%   |
>
> [1] Black, Kevin, et al. "$\pi_0 $: A Vision-Language-Action Flow Model for General Robot Control." arXiv preprint arXiv:2410.24164 (2024).
>
> [2] Kim, Sunghwan, et al. "PubChem substance and compound databases." Nucleic acids research 44.D1 (2016): D1202-D1213.
>
> [3] Zhang, Di, et al. "Chemllm: A chemical large language model." arXiv preprint arXiv:2402.06852 (2024).

---

### Official Review · Reviewer_3reT · 2025-07-18

**Rating:** 5
**Confidence:** 4

**Summary:**

This paper introduces a new simulation environment for training chemistry lab robots - both fixed 7DOF arms and arms mounted on a mobile base, based on Nvidia's Isaac Sim. They also introduce a benchmark set of tasks of progressively increasing difficulty within this environment. Finally, they present experimental results of running two baseline algorithms on trying to solve these tasks.

**Dataset Code Accessibility:**

Yes

**Dataset Code Comments:**

Code looks available; couldn't readily test as I'm on a mac and don't have a linux server handy at the moment.

**Ethical Considerations:**

No, there are no or only very minor ethics concerns

**Final Justification:**

I think this is a really cool environment - I think it's a solid foundation on which to build quite realistic and challenging tasks similar to the ones actually encountered by humans in real world labs. There are not many RL environments that can say the same. Also, lab work is a large field that is in great need for more automation, so I think it makes sense to have an RL environment specifically designed for this.

**Limitations Weaknesses:**

high level stuff:
- i wish the paper included biology lab equipment also instead of focusing on chemistry equipment only. the tasks and equipment feel kinda like stuff that you'd use in high school chemistry lab but not so much in the real world - i think far more money and time is spent on biology wetlab research than chemistry research nowadays. so stuff like pipettes/pipetting robots, well plates, various lidded tubes, centrifuges, etc would be cool to see. i've worked on robotic automation in a biology lab, integrating robot arms and pipetting robotics (eg OpenTrons style) to work together on tasks, which was quite effective. loading a centrifuge with a robot arm was super annoying and would be a cool task to see. as was screwing/unscrewing lids to falcon tubes. you might be interested to look at Emerald Cloud Lab, that's probably the best example i know of of a (partially) automated general biology wetlab.
- in general i think the environment was developed really nicely, but the task set seems pretty small, limited, and simple compared to what would be possible to build in this environment. i'd love to see more complex tasks especially in the level 4 category.
- i'm not too clear on how the environments are randomized for each episode of a task. i'd like more details explaining this. are the locations of the equipment randomized on each episode? i think they probably should be, at least slightly, because in real life there's always some positional error, or things getting shifted slightly, that can throw off the robotics. especially if the arm is on a mobile base, it'd be nice to model a bit of positional error. similarly for joint positions, you could consider adding a bit of noise to those inputs.

specific improvements:
- the intro could be condensed; i'd like to see more of the details of the environment, tasks, and experiment in the main paper instead. the section "Simulators for Embodied AI" doesn't seem super helpful - the pitch of this paper is "there isn't a simulator for lab robotics" and i think you can just say that and not have to compare deeply to other simulators. similarly table 1 could be dropped for the same reason.
- i'd like to see more videos of the environment and task and experimental results. maybe make a webpage for this paper with videos. i find that's the quickest way to get a feel for a new environment vs reading the paper and trying to imagine it.
- the inputs to the ML model need to be specified explicitly. just image? joint positions? joint velocities? in the navigation task, how does the robot know where the target object is?
- the outputs of the ML model also need specified explicitly. are you just directly predicting target full joint positions, or maybe torques/accelerations? if positions, how are these applied to the robot?
- i'm rather confused about the specific tasks. eg in appendix describing "pick" operation, you mention multiple phases. are these phases hardcoded or learned? if learned, why do you mention these explicit phases? or are these phases just used for generating the imitation learning data?
- in general i don't see it explained how the imitation learning data is generated. are these like human controlled trajectories? or from a hard coded policy? how are they randomized?
- you should say a bit more about where the cameras are located - ie are they fixed, or mounted on the robot?
- the code could use more detailed documentation, eg for how to add a new task. also it should be explicitly noted that this won't run on a M-series mac (because isaac sim does not support).
- you should explain why you chose these two algorithms to use as baselines, and what other things might be interesting to try.

nit:
- in appendix, "Asserts" should be assets. assets should be labeled with names.


I don't need all these responded to explicitly; these are just things I'd like to see improved in the paper. Expanding to biology is probably too big of a request for this current paper, but I'd love to see it in future iterations. My biggest request would be more varied+complex tasks. I think in general the simulator and the task set is the most valuable contribution of this type of work; the baseline experimental results will hopefully quickly become irrelevant if people do good research that builds on your environment.

**Strengths Contributions:**

- getting better at lab robotics would be super valuable and there are not good simulators for this that i know of, so i agree this type of work would be a really useful contribution. having both fixed arms and arms on mobile bases is nice too, both of these seem practical in real scenarios. i'd love to also see a pipetting robot and multi-robot cooperation in future versions of this work.
- the environment looks really nice. i haven't worked with isaac sim myself, but it looks quite nice and i think it's cool that we're starting to move more towards complex photorealistic environments.
- it's clear that a lot of effort went into modeling the various lab environments, and all the lab equipment that is available for building tasks from. this is really great, and will be useful building blocks for future work in this area regardless of how much of the rest of this work ends up getting built upon. for example, modeling equipment with interact-able doors and buttons is very nice - i've personally trained robot arms to work with this type of lab equipment and it's quite annoying to get right, so it's great to have these complexities directly in the sim.
- the LLM-based chemistry reaction simulation is interesting, although it doesn't seem to have really been actually used in the tasks?
- paper is generally well written

---

> ### Author Rebuttal · Authors · 2025-07-31
>
> Dear Reviewer 3reT,
>
> We sincerely thank the reviewer for the thoughtful, constructive, and encouraging feedback. We are truly grateful for your recognition of our simulator’s practical relevance, the value of modeling both fixed and mobile-arm robots, and the extensive effort invested in constructing detailed and interactable lab environments. We greatly appreciate your positive assessment of the simulator’s potential for broader impact in lab robotics and embodied AI.
>
> We fully agree with your key suggestions and have already incorporated several improvements based on your comments:
>
> > **Q1: Integration of the LLM-based chemical reaction module into laboratory tasks.**
>
> A1: Thank you for raising this valuable point. To better showcase the utility of the LLM-based chemical reasoning module, we have introduced a new Level‑4 task that explicitly requires reaction reasoning.
>
> Task description: Three small beakers are randomly placed on the table—two containing pure water and one containing a 1% starch solution. Another beaker holds a mixed solution of 0.05 mol/L potassium iodide (KI) and 0.01 mol/L iodine (I₂). The policy is tasked with sequentially pouring the contents of each small beaker into the beaker with the mixed solution, without prior knowledge of which beaker contains water or starch. Each time a solution is added, an LLM-based chemistry reaction engine is invoked to update the mixture's properties. When a color change reaction is detected, the robotic arm presses a button to indicate the end of the experiment.
>
> We evaluate the experiment using two success rates:
>
> 1. Action success rate: This measures whether all required actions are performed correctly—specifically, at least one pouring action followed by pressing the button to end the experiment—regardless of whether a color change occurs in the solution.
> 2. Strict success rate: This requires that, after a color change is observed, no additional solutions are poured into the beaker, and the button is successfully pressed to indicate the end of the experiment.
>
> Evaluation Results:
> Table 1: Performance comparison of different models on Level-4 Iodine Clock Reaction.
> |Model|Action success rate|Strict success rate|
> |-|---|---|
> |ACT|13%     | 4%  |
> |DP|6%      | 2%  |
>
> This task directly integrates the LLM-based engine to simulate and verify reaction feasibility. We view this as a foundation step towards broader experimental task domains, and continuously extend the benchmark with additional chemistry reasoning tasks in future releases.
>
> > **Q2: Extension for biology lab equipment and tasks.**
>
> A2: We sincerely thank the reviewer for this insightful suggestion. We fully agree that incorporating biology lab equipment, wetlab research tasks, and multi-robot collaboration is a valuable direction. We are committed to expanding LabUtopia accordingly, with the long-term goal of building a flexible, scalable simulator to support real-world automated science environments like Emerald Cloud Lab.
>
> > **Q3: More complex tasks especially in the level 4 category.**
>
> A3: Thanks for this valuable suggestion. We have added two Level‑4 long‑horizon tasks: 1) Iodine Clock Reaction (described in Q1), designed to evaluate the model’s ability in chemical reaction reasoning. 2) Laboratory bench cleaning task aimed at testing complex long-horizon arbitrary sequence operations:
>
> Task description:
> Laboratory bench cleaning task consists of three subtasks: close drying oven, pour liquid, and moving the beaker. These three tasks can be performed independently. During data collection, the execution order is randomly chosen. We trained both DP and ACT using 50 demonstrations for 100 epochs.
>
> Evaluation Results:
> Table 2: Performance comparison of different models on free action sequence task.
>
> |Model|Close drying oven|Pour liquid|Move the beaker|Overall success rate|
> |-|-|---|---|----|
> | ACT |48%|   36%     |40%     | 6%   |
> | DP  | 24%      |   32%   |      16%| 1%   |
>
> Furthermore, we will provide a template to facilitate users in extending experiments more easily, enabling both community‑driven extensions and broader task diversity.
>
> > **Q4: Explanation of the location and joint position randomization across episodes.**
>
> A4: Thank you for this suggestion. In our current setup, we typically fix the position of the robotic arm and randomize the object's position within a certain range.
>
> 1) For location randomization, object positions are randomized within a 15 cm × 15 cm region, with adjustments based on their sizes and workspace constraints. Nearly all manipulable objects undergo such perturbations to better reflect the positional uncertainty in real-world labs.
>
> 2) For joint position randomization, we agree that introducing noise to joint positions is a valuable improvement. Inspired by your suggestion, we incorporated this in our updated experiments by randomly adding Gaussian noise to the joint positions during training, and tested the results on the level-1 pick task, as shown in Tab. 3.
>
> We believe these forms of randomization are crucial for bridging the sim-to-real gap and we will clarify these details in the revised paper.
>
> Table 3: Performance with the added-noise strategy.
> | Pick Task |ACT | DP |
> |----|----|----|
> | non-noise |86.7|75.0|
> |added-noise|87.5|78.3|
>
> > **Q5: Conciseness of introduction.**
>
> A5: Thank you for this suggestion. We have streamlined the Introduction to emphasize the lack of a high-fidelity simulator for lab robotics and have expanded the benchmark description in the revision.
>
> > **Q6: Availability of videos and experimental demonstrations.**
>
> A6: We have provided an anonymous website in our abstract to include some videos of the tasks and plan to release comprehensive visualizations of full scenarios and tasks alongside the open-source code. Due to the conference policy, we cannot include images or external links at this stage (all materials will be made publicly available).
>
> > **Q7: Clarification of model inputs and outputs.**
>
> A7: Thank you for this suggestion. Our models use RGB images and joint positions as inputs. For navigation, we provide a location API that supplies the object’s 3D coordinates as part of the model input. The model's output is target joint positions, from which joint torques are computed via a PD controller to ensure precise trajectory execution. We will supply these details into Section 5.1 in the next version.
>
> > **Q8: Clarification of action phases and generation of imitation learning data.**
>
> A8: Sorry for the confusion. "multiple phases" refers to the sequential sub-phase in completing a single action. These phases are determined based on object-specific parameters such as position and volume, along with predefined hyperparameters, to ensure accurate execution. This hard-coded decomposition is used solely for collecting imitation learning data. The randomness stems from variations in object attributes and positions. We will clarify this process in the revision.
>
> > **Q9: Clarification of camera placement in experiments.**
>
> A9: In Section 5.1, we provide general principles for camera placement, following the original settings of policy models. To clarify further: all ACT and DP models were trained using two camera views — one directly overhead and one diagonally facing the robotic arm. For the open task, the camera facing the arm was mounted on the arm itself due to equipment obstructing the original position. In the added VLA experiments, we consistently used the camera mounted on the robot arm's wrist. We will revise the ‘Visual Input’ section in the revision.
>
> > **Q10: Clarification on code documentation and platform compatibility.**
>
> A10: Thank you for your valuable suggestion. We will provide comprehensive documentation in the open-source release, including a detailed template and step-by-step guide for adding new tasks, robots, and controllers. We also clarify the platform requirement: due to constraints of the NVIDIA Isaac Sim, the system currently requires an RTX-series GPU. We will explicitly state this requirement in the documentation.
>
> > **Q11: Selection of baseline algorithms.**
>
> A11: Thank you for the insightful question. We chose ACT and Diffusion Policy as baselines since they are representative transformer and diffusion‑based imitation learning methods and widely adopted in robotic manipulation research. These models serve as strong and well-established references for benchmarking task performance in our setting.
>
> In addition, recognizing the growing interest in Vision-Language-Action (VLA) models, we conducted supplementary experiments using the VLA-based model (Pi0[1]) on selected Level-3 generalization tasks (see Table 4). Specifically, we performed full fine-tuning based on the openpi0 base model, training each task for 30k steps on 8 GPUs. Interestingly, we observed two key findings:
>
> 1. Pretrained VLA models, after fine-tuning, do not always outperform models trained from scratch in our tasks.
> 2. VLA models show negligible performance degradation when evaluated on out-of-distribution visual inputs or novel materials.
>
> Table 4: Performance comparison on Level-3 Tasks (ID/OOD success rate %).
> | Task|Pi0 |ACT|DP|
> |-|-|-|-|
> | Pick     |83.3/85.8|81.7/71.7|53.3/41.7|
> | Press    |92.5/89.1|98.3/96.7|81.7/31.6|
> | Open Door|51.6/53.3|73.3/65.0|63.3/58.3|
> | Pour Liquid     |40.0/38.3|75.0/65.0|46.6/31.6|
> | Heater Beaker   |89.1/86.7|86.7/80.0|21.6/8.3 |
> | Transport Beaker|86.7/88.3|77.5/73.3|67.5/15.0|
>
> > **Q12: Typo correction and labeled assets.**
>
> A12: Thank you for pointing this out. We have corrected the typo “Asserts” to “Assets” in the appendix. Additionally, we have updated the asset descriptions with clear and consistent naming to enhance clarity and usability. We appreciate your attention to detail, which helps improve the quality of our work.
>
> [1] Black, Kevin, et al. "$\pi_0 $: A Vision-Language-Action Flow Model for General Robot Control." arXiv preprint arXiv:2410.24164 (2024).

---

> > ### Comment · Reviewer_3reT · 2025-08-05
> >
> > Thank you, this addressed the majority of my concerns with your work! I think this is a really cool environment and I'd love to see even more complex tasks added in the future - I think you've made a really solid foundation on which to build quite realistic and challenging tasks similar to the ones actually encountered by humans in real world labs. There are not many RL environments that can say the same.

---

> > > ### Author Response · Authors · 2025-08-06
> > >
> > > Thank you very much for your encouraging feedback and for recognizing the potential of our environment. We are delighted that our clarifications addressed most of your concerns.
> > >
> > > We greatly appreciate your suggestion regarding more complex tasks—expanding in this direction is definitely one of our future development goals. We will continue to increase the number of experiments and incorporate more complex tasks, making the environment even more useful for the research community.
> > >
> > > Best regards,
> > >
> > > The Authors

---

### Note · Authors · 2025-08-12

Dear SAC, AC, and reviewers,

We sincerely thank all of you for the time and effort. The valuable feedback from the reviewers has greatly helped us improve our work. Key enhancements include:
1. Expanded VLA experimentals: Added performance results of the VLA model on the benchmark.
2. Additional long-sequence experiments: Introduced the iodine clock and tabletop cleaning experiments, requiring a chemistry reaction engine and unordered task execution, further enriching benchmark diversity.
3. Protocol-following evaluation metrics:  Added evaluation protocols enforcing strict adherence to lab procedures, with success rates reported under these constraints.

During the rebuttal stage, two reviewers explicitly expressed their intention to raise their scores, and overall assessments remained highly positive. We deeply appreciate the reviewers’ professionalism and constructive engagement.

- Reviewer 3reT: We added new high-complexity tasks and corresponding evaluations. The reviewer views our environment as a solid foundation for building realistic and challenging tasks akin to those encountered in real-world laboratories.
- Reviewer Q5Ab: We supplemented the evaluation of the VLA model and chemical reasoning success rates to enhance the comprehensiveness of our benchmark. In addition, we introduced additional dimensions to the evaluation criteria to assess laboratory safety requirements.
- Reviewer mfSr: The reviewer noted that we added the incorporation of chemical reaction experiments and stricter success criteria. The reviewer believe these increased the appeal of the benchmark and intend to reflect into rating.
- Reviewer 385p: We presented LLM performance on chemical reaction tasks and the VLA experimental results, and believe we have addressed the reviewers’ concerns.
- Reviewer ibFk: We incorporated additional laboratory research tasks to broaden the coverage of science-centric reasoning scenarios and experimental operation scope.
- Reviewer 5G6a: Confirmed concerns were resolved, recognizing the additional LLM evaluations and new fluid experiment, and stated they would raise their final score.

In summary, all the feedback has been positive and expresses strong support for our work. We believe that the vast majority of concerns have been addressed. These changes will be included in the camera-ready version if accepted, as they significantly enhance the rigor and clarity of our work.

Thank you again for your time and effort!

Best, The Authors

---

### Decision · Program_Chairs · 2025-09-18

**Decision:**

Accept (poster)

**Comment:**

The reviewers agree that LabUtopia is a valuable contribution, providing a high-fidelity lab robotics simulator with a large curated asset library, realistic physics, and hierarchical tasks of increasing complexity. Strengths include its integration with Isaac Sim, modeling of fluids and chemical processes, photorealistic environments, and support for both fixed and mobile robot arms. The paper is well written, the benchmark well motivated, and the release of code and assets will make it a strong foundation for future work in lab automation.

Main limitations include the small and relatively simple task set, limited use of the chemical reaction simulation, and omission of biology lab equipment. The evaluation is narrow, testing only two imitation learning baselines without reinforcement learning or vision-language-action models. Reviewers also call for clearer details on task randomization, success criteria, model inputs/outputs, and imitation data generation, as well as richer demonstrations and videos. Despite these gaps, all reviewers rated it as a strong accept, seeing it as technically solid and impactful, with great potential for extension.

===== FINAL UPDATE FROM DB Track PCs ====

The final decision for this paper has been taken by the program chairs after consultation with the SACs. All Senior Area Chairs have ranked papers according to the feedback from the AC during the review process. We decided to leave the original meta-review to reflect the opinion of the AC in light of the initial discussions with reviewers and SAC.